# Causal Inference with Conditional Front-Door Adjustment and Identifiable Variational Autoencoder

**Ziqi Xu [1,2], Debo Cheng [1,✉], Jiuyong Li [1,✉], Jixue Liu [1], Lin Liu [1] & Kui Yu [3]**

University of South Australia [1]     Data61, CSIRO [2]     Hefei University of Technology [3]

`{firstname.lastname}@unisa.edu.au` [1]

`ziqi.xu@data61.csiro.au` [2]     `yukui@hfut.edu.cn` [3]

## Abstract

An essential and challenging problem in causal inference is causal effect estimation from observational data. The problem becomes more difficult with the presence of unobserved confounding variables. The front-door adjustment is an approach for dealing with unobserved confounding variables. However, the restriction for the standard front-door adjustment is difficult to satisfy in practice. In this paper, we relax some of the restrictions by proposing the concept of conditional front-door (CFD) adjustment and develop the theorem that guarantees the causal effect identifiability of CFD adjustment. By leveraging the ability of deep generative models, we propose CFDiVAE to learn the representation of the CFD adjustment variable directly from data with the identifiable Variational AutoEncoder and formally prove the model identifiability. Extensive experiments on synthetic datasets validate the effectiveness of CFDiVAE and its superiority over existing methods. The experiments also show that the performance of CFDiVAE is less sensitive to the causal strength of unobserved confounding variables. We further apply CFDiVAE to a real-world dataset to demonstrate its potential application.

## 1   Introduction

Estimating causal effects is a fundamental problem in many application areas. For example, policymakers need to know whether the implementation of a policy has a positive impact on the community (Athey, 2017; Tran et al., 2022), and medical researchers study the effects of treatments on patients (Petersen & van der Laan, 2014). Randomised Controlled Trials (RCTs) (Fisher, 1936) are considered the golden standard for estimating causal effects. However, RCTs are difficult to implement in many real-world cases due to ethical issues or high costs (Deaton & Cartwright, 2018). For example, it would be unethical to subject an individual to a condition (e.g., smoking) if the condition may have potentially negative consequences. Therefore, many methods have been developed to infer causal effects from observational data. Most of the methods assume no unobserved variables affecting both the treatment and outcome, i.e., the unconfoundedness assumption (Imbens & Rubin, 2015), and follow the back-door criterion (Pearl, 2009) to determine valid adjustment variable for unbiased estimation.

A graphical view of the typical cases in causal effect estimation is shown in Fig. 1. A simple case that satisfies the unconfoundedness assumption is illustrated in Fig. 1a. In this case, the causal effect can be unbiasedly estimated by back-door adjustment (Pearl, 2009). Fig. 1b, Fig. 1c and Fig. 1d show three cases where the unconfoundedness assumption is not satisfied. The IV (instrumental variable) approach has been extensively studied and commonly used to deal with the case shown in Fig. 1b. However, in practice, IV is not always available. In this case, if there exists a standard front-door adjustment variable, e.g., $Z_{\text{SFD}}$ as indicated in Fig. 1c, the standard front-door adjustment provides an effective approach to dealing with unobserved confounding variables.

However, the requirement for a valid standard front door adjustment variable is too strict, which hinders their practical application. In this paper, we aim to relax the requirement by considering a more practical setting as shown in Fig. 1d. Different from the standard front-door adjustment

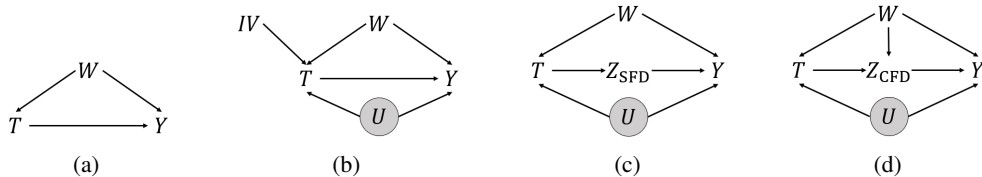

Figure 1: Typical cases in causal effect estimation. $T$ is the treatment; $Y$ is the outcome; $W$ is the observed confounding variable; $U$ is the unobserved confounding variable; $IV$ is the instrumental variable; $Z_{\text{SFD}}$ is the standard front-door adjustment variable; and $Z_{\text{CFD}}$ is the conditional front-door adjustment variable

setting in Fig. 1c, we allow the interaction between observed confounding variable ($W$) and the mediator ($Z_{\text{CFD}}$), and we call $Z_{\text{CFD}}$ a *conditional* front-door (CFD) adjustment variable. This is a more practical setting. For instance, referring to Fig. 1d, smoking ($T$) does not directly affect lung cancer development ($Y$) but mediated through tar in lungs ($Z_{\text{CFD}}$). For each patient, their other attributes such as age ($W$) can directly affect smoking, tar in lungs and lung cancer development. In this case, the standard front-door adjustment cannot be used since $Z_{\text{CFD}}$ is no longer a standard front-door adjustment variable because it does not meet the standard front-door criterion (Definition 3), since, there is an unblocked back-door path from $T$ to $Z_{\text{CFD}}$ ($T \leftarrow W \rightarrow Z_{\text{CFD}}$), and a back-door path from $Z_{\text{CFD}}$ to $Y$ ($Z_{\text{CFD}} \leftarrow W \rightarrow Y$) which is not blocked by $T$.

Additionally, it is unrealistic to assume that users always know a CFD adjustment variable in advance and thus it is desirable to find a CFD adjustment variable from observational data. In this paper, we propose a novel method, CFDiVAE, which is based on the identifiable VAE technique (Khemakhem et al., 2020) to learn the representation of a latent CFD variable from its proxy. We consider it is practical to assume the existence of proxies of a CFD adjustment variable. For instance, in the above example, the investigator may not observe tar in patients' lungs but they may observe the proxy variables, such as the results of patients' follow-up sputum tests and urine tests.

This paper advances the theory and practical use of causal inference in the presence of unobserved confounding variables through the following contributions:

- We address the challenge of estimating causal effects in the presence of unobserved confounding variables using the front-door adjustment. We relax some of the restrictions of the standard front-door adjustment by introducing conditional front-door (CFD) adjustment to enable more practical use of front-door adjustment. We also develop the theorem that guarantees the causal effect identifiability of CFD adjustment.

- We propose a novel method, CFDiVAE, that provides an effective data-driven method for dealing with unobserved confounding variables in causal effect estimation. We further provide the theoretical guarantee of the identifiability of the CFDiVAE.

- We evaluate the effectiveness of CFDiVAE on both synthetic and real-world datasets. Experiments with synthetic datasets show that CFDiVAE outperforms existing methods. Furthermore, we apply CFDiVAE to a real-world dataset to show the application scenarios and its potential.

## 2 PRELIMINARIES

We use a capital letter to represent a variable and a lowercase letter to represent its value. Due to page limitation, we provide the definitions of directed acyclic graph (DAG), causal path, non-causal path, markov condition, faithfulness, $d$-separation and $d$-connect in Appx. A.1.

This paper is focused on estimating the average treatment effect as defined below.

**Definition 1** (Average Treatment Effect (ATE)). *The average treatment effect of a treatment, denoted as $T$, on the outcome of interest, denoted as $Y$, is defined as $ATE = \mathbb{E}(Y \mid do(T = 1)) - \mathbb{E}(Y \mid do(T = 0))$, where $do()$ is the do-operator and $do(T = t)$ represents the manipulation of the treatment by setting its value to $t$ (Pearl, 2009).*

When the context is clear, we abbreviate $do(T = t)$ as $do()$. In order to allow the above $do()$ expressions to be recovered from data, Pearl formally defined causal effect identifiability (Pearl, 2009) (p.77) and proposed two well-known identification conditions, the back-door criterion and front-door criterion.

**Definition 2** (Back-Door Criterion (Pearl, 2009)). *A set of variables $Z_{BD}$ satisfies the back-door criterion relative to an ordered pair of variables $(T, Y)$ in a DAG $\mathcal{G}$ if: (1) no node in $Z_{BD}$ is a descendant of $T$; and (2) $Z_{BD}$ blocks every path between $T$ and $Y$ that contains an arrow into $T$.*

A back-door path is a non-causal path from $T$ to $Y$. They have been recognised as "back-door" paths because they flow backwards out of $T$, i.e., a back-door path points into $T$.

**Theorem 1** (Back-Door Adjustment (Pearl, 2009)). *If $Z_{BD}$ satisfies the back-door criterion relative to $(T, Y)$, then the causal effect of $T$ on $Y$ is identifiable and is given by the following back-door adjustment formula (Pearl, 2009):*

$$P(y|do(t)) = \sum_{z_{BD}} P(y \mid t, z_{BD})P(z_{BD}). \tag{1}$$

**Definition 3** (Front-Door Criterion (Pearl, 2009)). *A set of variables $Z_{SFD}$ is said to satisfy the (standard) front-door criterion relative to an ordered pair of variables $(T, Y)$ in a DAG $\mathcal{G}$ if: (1) $Z_{SFD}$ intercepts all directed paths from $T$ to $Y$; (2) there is no unblocked back-door path from $T$ to $Z_{SFD}$; and (3) all back-door paths from $Z_{SFD}$ to $Y$ are blocked by $T$.*

**Theorem 2** (Front-Door Adjustment (Pearl, 2009)). *If $Z_{SFD}$ satisfies the (standard) front-door criterion relative to $(T, Y)$, then the causal effect of $T$ on $Y$ is identifiable and is given by the following standard front-door adjustment formula (Pearl, 2009):*

$$P(y|do(t)) = \sum_{z_{SFD}, t'} P(y \mid t', z_{SFD})P(t')P(z_{SFD} \mid t), \tag{2}$$

*where $t'$ is a distinct realisation of treatment.*

## 3    CONDITIONAL FRONT-DOOR ADJUSTMENT

In this section, we present the definition of conditional front-door criterion and the theorem showing that the average causal effect of treatment $T$ on outcome $Y$ is identifiable via conditional front-door adjustment. The causal effect of $T$ on $Y$ is identifiable if the quantity $p(y \mid do(t))$ can be computed uniquely from any positive probability of the observed variables (Pearl, 2009). We formally define the conditional front-door criterion as follows:

**Definition 4** (Conditional Front-Door (CFD) Criterion). *A set of variables $Z_{CFD}$ is said to satisfy the conditional front-door criterion relative to an ordered pair of variables $(T, Y)$ in a DAG $\mathcal{G}$ if: (1) $Z_{CFD}$ intercepts all directed paths from $T$ to $Y$; (2) there exists a set of variables $W$, called the conditioning variables of $Z_{CFD}$, such that all back-door paths from $T$ to $Z_{CFD}$ are blocked by $W$; and (3) all back-door paths from $Z_{CFD}$ to $Y$ are blocked by $\{T\} \cup W$.*

Fig. 1d provides an illustration of CFD criterion, where $Z_{CFD}$ satisfies the criterion, and $W$ is the conditioning variable of $Z_{CFD}$. The following theorem provides the theoretical guarantee of the identifiability of the causal effect of $T$ on $Y$ via CFD adjustment and gives the adjustment formula.

**Theorem 3** (Conditional Front-Door (CFD) Adjustment). *If $Z_{CFD}$ satisfies the CFD criterion relative to $(T, Y)$, the causal effect of $T$ on $Y$ is identifiable and is given by the following CFD adjustment formula:*

$$P(y|do(t)) = \sum_{z_{CFD}, w, t'} P(y \mid t', z_{CFD}, w)P(t' \mid w)P(z_{CFD} \mid t, w)P(w), \tag{3}$$

*where $t'$ is a distinct realisation of treatment.*

Proof of the above theorem is provided in Appx. B.1.

In practice, we often do not have a given CFD adjustment variable, hence in the next section, we develop CFDiVAE to learn the representation of CFD adjustment variable using its proxies.

## 4    THE PROPOSED CFDiVAE METHOD

### 4.1   PROBLEM SETUP

We assume data is generated based on the DAG $\mathcal{G}$ in Fig. 2, where $T$ is the treatment variable, $Y$ is the outcome variable, $U$ is the unobserved confounding variable, $X_1, ..., X_n$ are the proxy of $Z_{CFD}$

$(n \geq 3)$, the latent CFD adjustment variable whose representation is to be learned and used for CFD adjustment, and $W_1, ..., W_m$ are the observed confounding variables and are the conditioning variables of $Z_{\text{CFD}}$ ($m \geq 1$). When the context is clear we omit the subscript.

In our problem setting, we assume that the observed confounding variable $W$ and the proxy variable $X$ are naturally separable. We believe this assumption is easy to satisfy in practice since $W$ is a pre-treatment variable (measured before treatment assignment) while $X$ is the proxy of the post-treatment variable, which is always collected after treatment assignment. For instance, with the example in the Introduction, as previously mentioned, $W$ can be a patient's age, and $X$ can be the results of some follow-up tests after the treatment has been applied, such as sputum and urine tests.

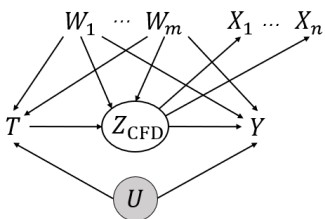

Figure 2: The DAG representing the data generation mechanism assumed in this paper.

To clarify, the latent variable (i.e., $Z_{\text{CFD}}$) refers to the variable that is not measured, but its proxies (i.e., $X_1, ..., X_n$) are measured. On the other hand, the unobserved confounding variable (i.e., $U$) is not measured and has no proxy. Latent variables and the existence of their proxies are commonly assumed by data-driven causal inference methods (Louizos et al., 2017; Zhang et al., 2021; Cheng et al., 2022) and it is a practical assumption. In addition to the previous example where follow-up medical test results can be a proxy for tar in lungs, another example would be in the case when we are not able to measure a person's economic status, so a common solution is to rely on the proxy variable such as postcode (Angrist & Pischke, 2009; Montgomery et al., 2000).

The efficacy of our method is contingent upon the quality of the proxy for the CFD adjustment variable, which is similar to most proxy based methods for causal inference. As discussed in (Goodman, 1974; Kruskal, 1976; Allman et al., 2009), the model can be identified if the latent variable has at least three independent proxies. Miao et al. (2018) build upon these assumptions, shifting the focus to causal effect estimation, posit that the causal effect could still be identified using only two independent proxies, even if the latent variable was not fully identified.

We summarise the problem setting as follows, including the assumptions made and the objective of the paper.

**Assumptions.** *There are always at least three independent proxy variables available for the latent CFD adjustment variable; the noisy level of the proxy variables should be small; and the distribution of the latent CFD adjustment variable should restrict to the exponential family distribution.*

**Objective.** *Given a joint probability distribution $P(X, W, T, Y)$ that is generated from the underlying DAG in Fig. 2 where $U$ and $Z_{\text{CFD}}$ are not measured. Suppose that $X = \{X_1, ..., X_n\}$ ($n \geq 3$) is the proxy of the latent variable $Z_{\text{CFD}}$ and $W = \{W_1, ..., W_m\}$ ($m \geq 1$) is the observed confounding variable. The goal of CFDiVAE is to learn the representation of $Z_{\text{CFD}}$.*

For the simplicity of notation and without causing confusion, in the rest of the paper, we use $Z_{\text{CFD}}$ to represent the learned representation of the latent variable $Z_{\text{CFD}}$ in Fig. 2, unless otherwise stated.

### 4.2 REPRESENTATION LEARNING

In this section, we introduce the details of CFDiVAE for learning $Z_{\text{CFD}}$. CFDiVAE learns a full generative model $p(X, Z_{\text{CFD}} \mid T, W) = p(X \mid Z_{\text{CFD}})p(Z_{\text{CFD}} \mid T, W)$ and an inference model $q(Z_{\text{CFD}} \mid T, W, X)$.

To guarantee the identifiability of CFDiVAE, we take $T$ and $W$ as additionally observed variables to approximate the prior $p(Z_{\text{CFD}} \mid T, W)$ (Khemakhem et al., 2020). Following existing VAE-based works in (Louizos et al., 2017; Zhang et al., 2021; Cheng et al., 2022), we assume the prior $p(Z_{\text{CFD}} \mid T, W)$ follows the Gaussian distribution.

In the inference model, we design the encoder $q(Z_{\text{CFD}} \mid T, W, X)$ that serves as the variational approximation of the posterior over the target representation, and the variational approximation of the posterior is defined as, $q(Z_{\text{CFD}} \mid T, W, X) = \prod_{j=1}^{D_{Z_{\text{CFD}}}} \mathcal{N}(\mu = \hat{\mu}_{Z_{\text{CFD}_j}}, \sigma^2 = \hat{\sigma}^2_{Z_{\text{CFD}_j}})$, where $\hat{\mu}_{Z_{\text{CFD}}}$ and $\hat{\sigma}^2_{Z_{\text{CFD}}}$ are the means and variances of the Gaussian distributions parameterised by the neural networks for $Z_{\text{CFD}}$.

The generative model for $X$ is defined as, $p(X \mid Z_{\text{CFD}}) = \prod_{j=1}^{D_X} \mathcal{N}(X_j \mid \mu = \hat{\mu}_{X_j}, \sigma^2 = \hat{\sigma}_{X_j}^2)$, where $\hat{\mu}_{X_j} = g(Z_{\text{CFD}_j})$; $\hat{\sigma}_{X_j}^2 = g(Z_{\text{CFD}_j})$, $D_X$ is the dimension of $X$, and $g(\cdot)$ is a neural network parameterised by its own parameters.

Then the evidence lower bound (ELBO) for the above inference and generative models is as follows:

$$\mathcal{M}_{\text{CFDiVAE}} = \mathbb{E}_q[\log p(X \mid Z_{\text{CFD}})] - D_{\text{KL}}[q(Z_{\text{CFD}} \mid T, W, X) \,||\, p(Z_{\text{CFD}} \mid T, W)], \qquad (4)$$

where $D_{\text{KL}}[\cdot||\cdot]$ is a KL divergence term.

## 4.3 MODEL IDENTIFIABILITY ANALYSIS

In this section, we provide the identifiability analysis of CFDiVAE. CFDiVAE is identifiable if the following implication holds.

$$\forall(\boldsymbol{\theta}, \boldsymbol{\theta}') : p_{\boldsymbol{\theta}}(X, Z_{\text{CFD}} \mid T, W) = p_{\boldsymbol{\theta}'}(X, Z_{\text{CFD}} \mid T, W) \implies \boldsymbol{\theta} = \boldsymbol{\theta}' \qquad (5)$$

Let $\boldsymbol{\theta} = (\text{f}, \text{S}, \boldsymbol{\lambda})$ be the parameters of the following conditional generative model:

$$p_{\boldsymbol{\theta}}(X, Z_{\text{CFD}} \mid T, W) = p_{\text{f}}(X \mid Z_{\text{CFD}}) p_{\text{S}, \boldsymbol{\lambda}}(Z_{\text{CFD}} \mid T, W), \qquad (6)$$

and we define:

$$p_{\text{f}}(X \mid Z_{\text{CFD}}) = p_{\boldsymbol{\varepsilon}}(X - \text{f}(Z_{\text{CFD}})). \qquad (7)$$

This means that the value of $X$ can be decomposed as $X = \text{f}(Z_{\text{CFD}}) + \boldsymbol{\varepsilon}$, where $\boldsymbol{\varepsilon}$ is an independent noise variable with probability density function $p_{\boldsymbol{\varepsilon}}(\boldsymbol{\varepsilon})$. However, our method also applies to non-noisy proxy variable and in this case $X = \text{f}(Z_{\text{CFD}})$.

The function f in Eq. 7 is injective and the $\boldsymbol{\varepsilon}$ should be small. For the prior $p_{\text{S}, \boldsymbol{\lambda}}(Z_{\text{CFD}} \mid T, W)$, we have the following assumption, i.e., conditionally factorial, where each element of $Z_{\text{CFD}}$ restricts to an exponential family distribution given $T$ and $W$.

The probability density function is given by:

$$p_{\text{S}, \boldsymbol{\lambda}}(Z_{\text{CFD}} \mid T, W) = \prod_i^{D_{Z_{\text{CFD}}}} \frac{Q_i(Z_{\text{CFD}_i})}{Z_i(T, W)} exp \left[ \sum_{j=1}^{k} S_{i,j}(Z_{\text{CFD}_i}) \lambda_{i,j}(T, W) \right], \qquad (8)$$

where $Q_i$ is the base measure, $Z_i(T, W)$ is the normalising constant and $S_i = (S_{i,1}, ..., S_{i,k})$ are sufficient statistics and $\boldsymbol{\lambda}(T, W) = (\lambda_{i,1}(T, W), ..., \lambda_{i,k}(T, W))$ are the corresponding parameters depending on $T$ and $W$, and $k$ is the dimension of each sufficient statistic.

Following the work in (Khemakhem et al., 2020), let $X \in \mathbb{R}^d$ and $Z_{\text{CFD}} \in \mathbb{R}^n$ ($n \leq d$), we have the following theorem about the identifiability of CFDiVAE.

**Theorem 4.** *Assume that the observational data are generated according to Eq. 6-Eq. 8 with parameters $\boldsymbol{\theta} = (\text{f}, \text{S}, \boldsymbol{\lambda})$ and the following hold: (1) The function f in Eq. 7 is injective. (2) The set $\{X \in \mathcal{X} \mid \varphi_{\text{f}}(X) = 0\}$ has measure zero, where $\varphi_{\boldsymbol{\varepsilon}}$ is the characteristic function of the density $p_{\boldsymbol{\varepsilon}}$ defined in Eq. 7. (3) The sufficient statistics $S_{i,j}$ in Eq. 8 are differentiable almost everywhere, and $(S_{i,j})_{1 \leq j \leq k}$ are linearly independent on any subset of $\mathcal{X}$ of measure greater than zero. (4) There exists $nk + 1$ distinct points $(T, W)_0, ..., (T, W)_{nk}$ such that the matrix $\boldsymbol{L} = (\boldsymbol{\lambda}(T_1, W_1) - \boldsymbol{\lambda}(T_0, W_0), ..., \boldsymbol{\lambda}(T_{nk}, W_{nk}) - \boldsymbol{\lambda}(T_0, W_0))$ of size $nk \times nk$ is invertible. Then the parameters $\boldsymbol{\theta} = (\text{f}, \text{S}, \boldsymbol{\lambda})$ are $\sim_A$-identifiable.*

This theorem guarantees the identifiability of the generative model in Eq. 6. Proof of the theorem is provided in Appx. B.2 and more related definitions are available in Appx. A.2. We also provide an analysis in Appx. B.3 to show the validity of the learned (transformed) CFD adjustment variable.

## 5 ATE ESTIMATION

After learning $Z_{\text{CFD}}$, we can obtain unbiased estimation of the ATE by using the CFD adjustment. Due to the page limit, we show how this is done with discrete data in Algorithm 1 in Appx. C. For continuous data, the methods would vary depending on the data generation models. We provide the basic solution for data generation under a linear model in Appx. C. For data generated under a nonlinear model, we will explore this as a future work.

# 6 EXPERIMENTS

In this section, we first demonstrate the correctness of representation learning. Then, we compare the performance of CFDiVAE with the benchmark methods for estimating causal effects and validate that CFDiVAE can unbiasedly estimate the causal effects and its performance is not sensitive to the change of the causal strength of the unobserved confounding variable. We also show its feasibility when the dimension of the learned representation is mismatched with the dimension of the ground truth CFD adjustment variable. Finally, we apply CFDiVAE to a real-world dataset and demonstrate its potential application. We also provide an additional experiment on the analysis of model identifiability in Appx. D.3. The source code is available in the Supplementary Material.

## 6.1 EXPERIMENT SETUP

We compare CFDiVAE with a number of benchmark methods, including traditional and VAE based causal effect estimation methods. Note that all those methods are based on the back-door adjustment. FindFDSet and ListFDSets (Jeong et al., 2022; Wienöbst et al., 2022) are used in the experiments since they are front-door adjustment based methods. FindFDSet aims to find the standard front-door adjustment variable in a given DAG, while ListFDSets is an extension and it can enumerate the standard front-door adjustment sets.

Table 1: Methods for comparison.

| Name | Open-Source |
|---|---|
| LinearDRL (Chernozhukov et al., 2018) | EconML |
| CausalForest (Wager & Athey, 2018) | EconML |
| ForestDRL (Athey et al., 2019) | EconML |
| XLearner (Künzel et al., 2019) | EconML |
| KernelDML (Nie & Wager, 2021) | EconML |
| CEVAE (Louizos et al., 2017) | GitHub |
| TEDVAE (Zhang et al., 2021) | GitHub |
| FindFDSet (Jeong et al., 2022) | GitHub |
| ListFDSets (Jeong et al., 2022) | GitHub |

The implementations of CEVAE and TEDVAE are retrieved from the authors' GitHub, the implementations of other methods are from EconML (Keith Battocchi, 2019), and the implementations of FindFDSet and FindFDSet are retrieved from the authors' GitHub as shown in Table 1. A detailed description of the comparison methods is shown in Appx. D.1.

For evaluating the performance of CFDiVAE and the benchmark methods, we use the Estimation Bias $|(\hat{\beta} - \beta)/\beta| \times 100\%$ as the metric, where $\hat{\beta}$ is the estimated ATE and $\beta$ is the ground truth.

The evaluation of estimated causal effects with unobserved confounding variables relies on synthetic datasets since no ground truth causal effects available for real-world datasets (Louizos et al., 2017; Zhang et al., 2021; Cheng et al., 2022). Synthetic datasets used in the evaluation are generated based on the causal graph (mechanism) shown in Fig 2. More details on data generation are provided in the Supplementary Material. To avoid the bias brought by the data generation process, we repeatedly generate 30 datasets with a range of sample sizes (denoted as N), including 0.5k, 1k, 2k, 4k, 6k, 8k, 10k and 20k. For each method, we report the average (mean) estimation bias over the 30 datasets, together with the standard deviation.

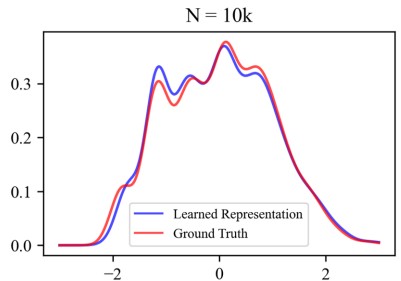

Figure 3: Probability Density Functions of the ground truth and the learned representation, where the horizontal axis represents the value and the vertical axis represents the density.

## 6.2 CORRECTNESS OF THE LEARNED REPRESENTATION

In this section, we conduct experiments to validate the correctness of the learned representation. Since we use synthetic datasets, we know the ground truth of the CFD adjustment variable. To evaluate the correctness of the representations learned by CFDiVAE, we compare the probability distribution of the learned representation against the distribution of the corresponding ground truth CFD adjustment variable. Due to the page limit, we only show the result of N=10k. As shown in Fig. 3, the distribution of the learned representation is close to the distribution of the ground truth, which indicates that CFDiVAE can learn accurate representation of the CFD adjustment variable. More results are reported in Appx. D.2.

Table 2: The estimation bias (%) of CFDiVAE and comparison methods under different $N$ values.

|  | 0.5k | 1k | 2k | 4k | 6k | 8k | 10k | 20k |
|---|---|---|---|---|---|---|---|---|
| LinearDRL | 21.90 ± 5.13 | 21.56 ± 3.82 | 21.47 ± 3.28 | 21.82 ± 2.08 | 21.59 ± 1.78 | 21.88 ± 1.41 | 21.89 ± 1.31 | 21.38 ± 0.90 |
| CausalForest | 21.87 ± 5.55 | 21.33 ± 4.28 | 21.39 ± 3.62 | 21.85 ± 1.98 | 21.63 ± 1.80 | 21.88 ± 1.33 | 21.94 ± 1.23 | 21.36 ± 0.99 |
| ForestDRL | 21.90 ± 4.95 | 21.58 ± 3.69 | 21.36 ± 3.38 | 21.79 ± 2.04 | 21.54 ± 1.80 | 21.88 ± 1.38 | 21.89 ± 1.28 | 21.41 ± 0.89 |
| XLearn | 21.92 ± 5.14 | 21.65 ± 3.55 | 21.35 ± 3.36 | 21.83 ± 2.04 | 21.59 ± 1.78 | 21.86 ± 1.39 | 21.88 ± 1.30 | 21.39 ± 0.90 |
| KernelDML | 19.57 ± 5.38 | 19.63 ± 3.83 | 19.79 ± 3.56 | 20.38 ± 2.04 | 20.24 ± 1.75 | 20.59 ± 1.39 | 20.64 ± 1.25 | 20.27 ± 0.94 |
| CEVAE | 102.63 ± 2.83 | 104.31 ± 7.82 | 101.42 ± 20.50 | 31.05 ± 4.95 | 26.93 ± 5.04 | 23.97 ± 6.05 | 21.29 ± 6.81 | 28.83 ± 4.72 |
| TEDVAE | 98.91 ± 17.37 | 70.73 ± 16.94 | 26.67 ± 3.58 | 24.63 ± 2.28 | 22.84 ± 1.85 | 22.67 ± 1.61 | 22.63 ± 1.23 | 21.84 ± 0.98 |
| FindFDSet | 15.79 ± 8.96 | 16.15 ± 7.77 | 16.33 ± 6.10 | 16.58 ± 3.11 | 15.64 ± 2.40 | 16.44 ± 2.36 | 16.49 ± 2.65 | 16.53 ± 1.63 |
| ListFDSet | 15.79 ± 8.82 | 15.97 ± 7.72 | 16.20 ± 6.10 | 16.43 ± 3.10 | 15.52 ± 2.42 | 16.31 ± 2.32 | 16.36 ± 2.60 | 16.39 ± 1.60 |
| CFDiVAE | 86.29 ± 6.21 | 39.72 ± 31.47 | 8.87 ± 10.68 | 4.57 ± 3.03 | 2.58 ± 1.96 | 2.32 ± 1.47 | 2.97 ± 2.09 | 2.14 ± 3.38 |

## 6.3 PERFORMANCE OF ATE ESTIMATION

In this section, we evaluate the performance of CFDiVAE in estimating the ATE compared with benchmark methods. As demonstrated in Table 2, CFDiVAE outperforms all other comparison methods when the sample size is 2k or larger. This outcome is anticipated. All traditional and VAE-based comparison methods employ back-door adjustment to estimate ATE, utilizing $W$ as the back-door adjustment variable. The estimation bias for comparison methods arises from the unobserved confounding variable $U$. To achieve unbiased estimation based on back-door adjustment, all back-door paths between $T$ and $Y$ must be blocked, which is unattainable due to the unobservable nature of $U$. Our proposed method, CFDiVAE, avoids the limitations of back-door adjustment.

For this set of experiments, the PC algorithm implemented in the causal discovery toolbox TETRAD (Ramsey et al., 2018) is used to generate DAGs as inputs for FindFDSet and ListFD-Sets. These two methods achieve better performance compared to back-door adjustment methods. However, there is still some bias resulting from causal structure learning and errors introduced by proxy variables. We note that our method performs well only when the sample size is sufficiently large, which is also observed in other VAE methods.

## 6.4 IMPACT OF THE CAUSAL STRENGTH OF UNOBSERVED CONFOUNDING VARIABLE

We also conduct experiments to verify the effectiveness of CFDiVAE with respect to different causal strengths of the unobserved confounding variable. For this set of experiments, the causal strength is varied by adjusting the coefficient of the path $U \to Y$. The sample size for this experiment is fixed at 10k. We multiply a scaling factor to the coefficient (i.e., $\beta_{U,Y}$) to realise the different causal strength levels of the unobserved confounding variable. For example, $0.0$ means that there is no unobserved confounding variable, and $2.0$ means that the coefficient doubles the original value. The range of the scaling factor is set as $[0.0, 2.0]$ and the step increment is set as $0.2$.

The results are shown in Fig. 4. When the causal strength is zero, i.e., no unobserved confounding variable, the back-door adjustment based comparison methods each achieve their own best performance since in this case, all confounding variables are observed and their performance is solely determined by their capabilities in correctly identifying or learning the correct back-door adjustment variable. With the increase of causal strength, there is a clear downward trend in the performance of the back-door adjustment based comparison methods, indicating that the back-door adjustment cannot handle unobserved confounding variables. In contrast, the front-door based methods retain an unchanged trend in estimation performance. CFDiVAE achieves and maintains an estimation bias of around $3\%$. The result is expected as CFDiVAE is based on the CFD adjustment, which can cope with unobserved confounding variables.

## 6.5 SENSITIVITY TO REPRESENTATION DIMENSION

In real-world applications, it is a common situation that the dimension set for the representation does not match the dimension of the ground truth CFD adjustment variable. In this section, we analyse the sensitivity of CFDiVAE to the representation dimension. In the following, $D_L$ represents the dimension of the learned representation, $D_R$ represents the dimension of the ground truth CFD adjustment variable, and $D_X$ represents the dimension of the proxy variable. We apply CFDiVAE with various dimension settings, i.e., $D_R \in \{2, 4, 8\}$. The results are shown in Table 3. We see that CFDiVAE achieves its best performance in most sample sizes when $D_L = D_X$, which is similar to the observation shown in (Wu & Fukumizu, 2022). When $D_L \neq D_R$ and $D_L \neq D_X$, the performance

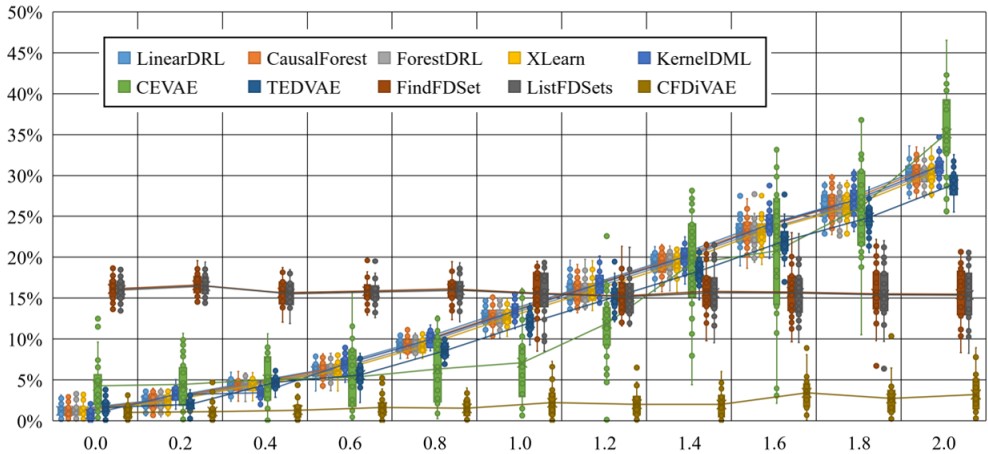

Figure 4: Results with different scaling factor, where the horizontal axis represents the scaling factor and the vertical axis represents the estimation bias (%).

Table 3: The estimation bias (%) of CFDiVAE to dimension mismatch on different N values. The best results are shown in **boldface**.

| $D_L$-$D_R$-$D_X$ | 0.5k | 1k | 2k | 4k | 6k | 8k | 10k | 20k |
|---|---|---|---|---|---|---|---|---|
| 1-2-4 | 82.31 ± 8.83 | **11.99 ± 5.98** | 10.7 ± 17.07 | 9.52 ± 3.08 | 9.54 ± 2.34 | 9.86 ± 2.54 | 10.35 ± 4.25 | 9.88 ± 1.36 |
| 2-2-4 | 78.16 ± 4.99 | 12.85 ± 10.96 | **6.90 ± 5.88** | 8.83 ± 6.02 | 5.94 ± 4.22 | 5.46 ± 3.62 | 5.37 ± 6.82 | 4.16 ± 8.90 |
| 4-2-4 | **76.69 ± 4.44** | 23.21 ± 15.38 | 12.31 ± 19.33 | 7.98 ± 6.98 | 5.73 ± 5.45 | 7.45 ± 3.36 | 3.90 ± 4.87 | 1.77 ± 1.08 |
| 1-4-8 | 79.94 ± 8.98 | 22.12 ± 18.63 | 12.09 ± 4.62 | 13.73 ± 3.58 | 14.24 ± 3.43 | 15.07 ± 2.86 | 14.33 ± 2.64 | 14.83 ± 1.74 |
| 2-4-8 | 74.31 ± 6.90 | **16.38 ± 8.40** | **9.49 ± 5.02** | 11.54 ± 3.75 | 9.84 ± 3.15 | 8.19 ± 4.86 | 8.43 ± 6.85 | 6.10 ± 1.83 |
| 4-4-8 | 73.16 ± 5.70 | 19.04 ± 11.12 | 12.89 ± 16.47 | **9.90 ± 5.15** | 8.74 ± 5.69 | 6.78 ± 3.92 | 4.50 ± 2.70 | 4.45 ± 1.75 |
| 8-4-8 | **69.66 ± 8.20** | 31.97 ± 12.11 | 38.96 ± 12.17 | 11.73 ± 4.09 | **8.05 ± 6.67** | **6.07 ± 3.75** | 4.42 ± 2.46 | **4.44 ± 2.00** |
| 1-8-16 | 75.38 ± 12.60 | 27.92 ± 22.61 | 15.86 ± 6.77 | 14.46 ± 17.05 | 15.18 ± 4.85 | 16.62 ± 3.73 | 16.64 ± 3.34 | 17.65 ± 2.29 |
| 4-8-16 | 72.33 ± 11.25 | **19.45 ± 13.21** | **12.89 ± 10.46** | 11.71 ± 12.01 | 12.28 ± 5.88 | 10.36 ± 5.37 | 9.26 ± 5.88 | 7.47 ± 3.04 |
| 8-8-16 | 63.95 ± 11.41 | 27.47 ± 13.96 | 29.00 ± 26.66 | 11.01 ± 11.10 | 10.00 ± 7.61 | 9.00 ± 6.27 | 6.69 ± 4.29 | 7.42 ± 3.14 |
| 16-8-16 | **56.84 ± 11.64** | 22.67 ± 12.68 | 17.15 ± 19.49 | **10.60 ± 9.62** | **7.86 ± 6.56** | **8.28 ± 5.75** | **6.23 ± 4.15** | **5.30 ± 2.99** |

of CFDiVAE can also maintain at an acceptable level. In all cases, the performance of CFDiVAE is superior to the comparison methods (Appx. D.4 shows more results). Hence, when the dimension of the ground truth CFD adjustment variable is not accessible, we can safely set $D_L = D_X$.

## 6.6 CASE STUDY ON A REAL-WORLD DATASET

In this section, we apply CFDiVAE to detect discrimination on the real-world dataset, Adult. The dataset is retrieved from the UCI repository (Dua & Graff, 2017) and it contains 11 attributes about personal, educational and economic information for 48842 individuals. We use the sensitive attribute sex as $T$, income as $Y$, and age, race and native_country as $W$.

With causality-based discrimination detection, we consider that there is direct discrimination if the sensitive attribute has a large enough direct causal effect on the outcome (above a given threshold $\tau$), and there is indirect discrimination if the sensitive attribute has a large enough indirect causal effect on the outcome and the mediator is also a sensitive attribute (Zhang et al., 2018).

Following the causal network in Fig. 7 in Appx. D.5, the green path represents the direct path from sex to income, and the blue paths represent the indirect paths passing through marital_status. The discrimination threshold $\tau$ is set as 0.05. By computing the path-specific effects, Zhang et al. (2017a) estimated that direct effect = 0.025 and indirect effect = 0.175, which indicated no direct discrimination but significant indirect discrimination.

We aim to estimate the causal effect of sex on income using representation learning and conditional front-door adjustment. We simplify the causal network (Fig. 7 in Appx. D.5) to fit our method with a latent stereotype as shown in Fig. 8 in Appx. D.5. The discrimination is not a direct result of sex, but a direct result of the stereotype. We assume the proxy of the stereotype is accessible, and they are marital_status, relationships, edu_level, hours_per_week, occupation, workclass, in this example. Stereotype is not a standard front-door adjustment variable because the causal path from stereotype to income is not blocked by sex. However, the stereotype is a CFD adjustment variable, since there

are no back-door paths from sex to stereotype and all back-door paths from stereotype to income are blocked by age, race and native_country (adding sex into this adjustment set will not invalidate the result). By using CFDiVAE, we obtain that ATE = 0.176, which is consistent with the previous estimate (0.175). The direct effect is ignored since it is very small.

# 7 RELATED WORK

Over the past few decades, researchers have proposed many methods for estimating causal effects from observational data. These methods generally fall into three categories: methods based on back-door adjustment, instrumental variables (IVs) and front-door adjustment, respectively.

Methods based on back-door adjustment are the most widely used, and most of these methods need to assume that all confounding variables are observed. For example, several tree-based methods (Athey & Imbens, 2016; Su et al., 2009; Zhang et al., 2017b) have been designed to estimate causal effects by designing specific splitting criteria; meta-learning (Künzel et al., 2019) has also been proposed to utilise existing machine learning algorithms for causal effect estimation. Recently, methods using deep learning techniques to predict causal effects have received widespread attention. For example, CEVAE (Louizos et al., 2017) combines representation learning and VAE to estimate causal effects; TEDVAE (Zhang et al., 2021) improves on CEVAE and decouples the learned representations to achieve more accurate estimation; $\beta$-Intact-VAE (Wu & Fukumizu, 2022) is based on the identifiable VAE technique which models a prognostic score with a latent variable to identify and estimate individualised treatment effects in situations of limited overlap.

Methods based on IVs have also received a lot of attention. Most IV based methods require users to nominate a valid IV, such as the generalised method of moments (GMM) (Bennett et al., 2019), kernel-IV regression (Singh et al., 2019) and deep learning based method (Hartford et al., 2017). When there are no nominated IVs in the data, some data-driven methods are developed to find (Yuan et al., 2022) or synthesise (Burgess & Thompson, 2013; Kuang et al., 2020) an IV or eliminate the influence of invalid IVs by using statistical strategies (Guo et al., 2018; Hartford et al., 2021).

Front-door adjustment approach is rarely studied in the literature. There are only a few methods for finding appropriate adjustment sets by following the standard front-door criterion (Jeong et al., 2022; Wienöbst et al., 2022). These methods require a given DAG and aim to find and enumerate possible standard front-door adjustment variables in the DAG.

The methods based on back-door adjustment cannot handle unobserved confounding variables. IV based methods can cope with unobserved confounding variables, but the availability of known IVs is itself a strong assumption. Existing front-door adjustment methods are restricted to the standard front-door adjustment variable. We relax the restriction of standard front-door adjustment and develop CFDiVAE to learn a CFD adjustment variable for unbiased ATE estimation in the presence of unobserved confounding variables.

# 8 CONCLUSION

**Summary of Contributions.** In this work, we proposed the conditional front-door adjustment, which is less restrictive than the standard front-door adjustment and proved that the average causal effect is identifiable via the proposed conditional front-door adjustment. Our proposed method CFDiVAE leverages the identifiable VAE technique to learn the representation of the conditional front-door adjustment variable from data directly, and we have shown that the identifiability of the learned representation is theoretically guaranteed. Extensive experiments have demonstrated that CFDiVAE outperforms the benchmark methods. We have also shown that CFDiVAE is insensitive to the causal strength of the unobserved confounding variable. Furthermore, the case study has suggested the potential of CFDiVAE for real-world applications.

**Limitations & Future Works.** The efficacy of the CFD adjustment and the CFDiVAE method rely on a set of assumptions. We acknowledge that such assumptions may not hold in some scenarios. Notably, the robustness of CFDiVAE is largely dependent on the availability of a substantial sample size. Presently, our solution is only for continuous data within linear models and discrete data. In the future, our endeavors will be directed toward relaxing these assumptions. Additionally, we aim to evolve the CFD adjustment technique for application to continuous data with nonlinear models.

ACKNOWLEDGMENTS

We wish to acknowledge the support from the Australian Research Council Discovery Project 200101210. Ziqi Xu is supported by the University Presidents Scholarship (UPS) of the University of South Australia. Kui Yu is supported by the National Natural Science Foundation of China under Grant 62376087. We are grateful for the valuable feedback provided by all anonymous reviewers.

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

# A BACKGROUND

## A.1 CAUSALITY

Let $\mathcal{G} = (\mathbf{V}, \mathbf{E})$ be a directed acyclic graph (DAG), where $\mathbf{V}$ is the set of nodes and $\mathbf{E}$ is the set of edges between the nodes. A path $\pi$ between nodes $V_1$ and $V_n$ comprises a sequence of distinct nodes $< V_1, \ldots, V_n >$ with every pair of successive nodes being adjacent. A node $V$ lies on the path $\pi$ if $V$ belongs to the sequence $< V_1, \ldots, V_n >$.

A path $\pi$ is causal if all edges along it are all in the same direction such as $V_1 \rightarrow \ldots \rightarrow V_n$. A path that is not causal is referred to as a non-causal path.

**Assumption 1** (Markov Condition (Pearl, 2009)). *Given a DAG $\mathcal{G} = (\mathbf{V}, \mathbf{E})$ and $P(\mathbf{V})$, the joint probability distribution of $\mathbf{V}$, $\mathcal{G}$ satisfies the Markov Condition if $\forall V_i \in \mathbf{V}$, $V_i$ is probabilistically independent of all of its non-descendants, given $Pa(V_i)$, the set of all parent nodes of $V_i$.*

**Assumption 2** (Faithfulness (Spirtes et al., 2000)). *A DAG $\mathcal{G} = (\mathbf{V}, \mathbf{E})$ is faithful to $P(\mathbf{V})$ iff every conditional independence present in $P(\mathbf{V})$ is entailed by $\mathcal{G}$ and satisfies the Markov Condition. $P(\mathbf{V})$ is faithful to $\mathcal{G}$ iff there exists $\mathcal{G}$ which is faithful to $P(\mathbf{V})$.*

When the Markov condition and faithfulness assumption are satisfied, we can use $d$-separation to read the conditional independence between variables entailed in the DAG $\mathcal{G}$.

**Definition 5** ($d$-separation (Pearl, 2009)). *A path $\pi$ in a DAG is said to be d-separated (or blocked) by a set of nodes $Z$ iff (1) the path $\pi$ contains a chain $V_i \rightarrow V_k \rightarrow V_j$ or a fork $V_i \leftarrow V_k \rightarrow V_j$ such that the middle node $V_k$ is in $Z$, or (2) the path $\pi$ contains an inverted fork (or collider) $V_i \rightarrow V_k \leftarrow V_j$ such that $V_k$ is not in $Z$ and no descendant of $V_k$ is in $Z$.*

Let $\mathcal{G} = (\mathbf{V}, \mathbf{E})$ be a DAG, and $P(V)$ is the probability distribution over $V$. In the DAG $\mathcal{G}$, a set of nodes $Z$ is said to $d$-separate $V_i$ and $V_j$ if and only if $Z$ blocks every path between $V_i$ to $V_j$; otherwise, a set of nodes $Z$ is said to $d$-connect $V_i$ and $V_j$. When the Markov Condition and Faithfulness assumption are satisfied by $\mathcal{G}$ and $P(V)$, $(V_i \perp\!\!\!\perp V_j \mid Z)$ if $Z$ $d$-separates $V_i$ and $V_j$, and $(V_i \not\perp\!\!\!\perp V_j \mid Z)$ if $Z$ $d$-connects $V_i$ and $V_j$.

**Theorem 5** (Rules of $do$-Calculus (Pearl, 2009)). *Let $\mathcal{G}$ be the DAG associated with a causal model, and let $P(\cdot)$ stand for the probability distribution induced by that model. For any disjoint subsets of variables $T, Y, Z,$ and $W$, we have the following rules.*

*Rule 1. (Insertion/deletion of observations):*
$$P(y \mid do(t), z, w) = P(y \mid do(t), w), \text{ if } (Y \perp\!\!\!\perp Z \mid T, W) \text{ in } \mathcal{G}_{\overline{T}}.$$

*Rule 2. (Action/observation exchange):*
$$P(y \mid do(t), do(z), w) = P(y \mid do(t), z, w), \text{ if } (Y \perp\!\!\!\perp Z \mid T, W) \text{ in } \mathcal{G}_{\overline{T}\underline{Z}}.$$

*Rule 3. (Insertion/deletion of actions):*
$$P(y \mid do(t), do(z), w) = P(y \mid do(t), w), \text{ if } (Y \perp\!\!\!\perp Z \mid T, W) \text{ in } \mathcal{G}_{\overline{T}, \overline{Z(W)}},$$

*where $Z(W)$ is the nodes in $Z$ that are not ancestors of any node in $W$ in $\mathcal{G}_{\overline{T}}$.*

## A.2 MODEL IDENTIFIABILITY

We define two equivalence relations on the set of parameters $\Theta$.

**Definition 6.** *Let $\sim$ be the equivalence relation on $\Theta$ defined as follows:*

$$(\mathrm{f}, \mathrm{S}, \boldsymbol{\lambda}) \sim_{\boldsymbol{A}} (\widetilde{\mathrm{f}}, \widetilde{\mathrm{S}}, \widetilde{\boldsymbol{\lambda}}) \Leftrightarrow$$
$$\exists \boldsymbol{A}, \mathrm{c} \mid \mathrm{S}(\mathrm{f}^{-1}(\boldsymbol{x})) = \boldsymbol{A}\widetilde{\mathrm{S}}(\widetilde{\mathrm{f}}^{-1}(\boldsymbol{x})) + \mathrm{c}, \forall \boldsymbol{x} \in \mathcal{X}, \tag{9}$$

*where $(\widetilde{\mathrm{f}}, \widetilde{\mathrm{S}}, \widetilde{\boldsymbol{\lambda}})$ are the parameters obtained from some learning algorithm that perfectly approximates the marginal distribution of the observations, $\boldsymbol{A}$ is an invertible $nk \times nk$ matrix, $\mathrm{c}$ is a vector, and $\mathcal{X}$ is the domain of $X$.*

# B PROOFS

## B.1 PROOF OF THEOREM 3

*Proof.* We compute $P(y \mid do(t))$ by using Theorem 5 under the DAG $\mathcal{G}$ in Fig. 2. Fig. 5 shows the subgraphs that are needed for the derivations in the following.

$P(y \mid do(t))$ can be expanded as:

$$P(y \mid do(t)) = \sum_{z_{\text{CFD}}} P(z_{\text{CFD}} \mid do(t))P(y \mid z_{\text{CFD}}, do(t)) \tag{10}$$

We first compute $P(y \mid z_{\text{CFD}}, do(t))$, which can be expanded as follow:

$$P(y \mid z_{\text{CFD}}, do(t)) = \sum_{w} P(y \mid do(t), z_{\text{CFD}}, w)P(w \mid do(t), z_{\text{CFD}}) \tag{11}$$

The first part: $P(y \mid do(t), z_{\text{CFD}}, w) = P(y \mid do(t), do(z_{\text{CFD}}), w)$,
  since $(Y \perp\!\!\!\perp Z_{\text{CFD}} \mid T, W)$ in $\mathcal{G}_{\overline{T}\underline{Z_{\text{CFD}}}}$ (Rule 2 in Theorem 5)

$P(y \mid do(t), do(z_{\text{CFD}}), w) = P(y \mid do(Z_{\text{CFD}}), w)$,
  since $(Y \perp\!\!\!\perp T \mid Z_{\text{CFD}}, W)$ in $\mathcal{G}_{\overline{Z_{\text{CFD}}T(W)}}$ (Rule 3 in Theorem 5)

$P(y \mid do(z_{\text{CFD}}), w) = \sum_{t'} P(y \mid do(z_{\text{CFD}}), t', w)P(t' \mid do(z_{\text{CFD}}), w)$

$P(y \mid do(z_{\text{CFD}}), t', w) = P(y \mid z_{\text{CFD}}, t', w)$,

  since $(Y \perp\!\!\!\perp Z_{\text{CFD}} \mid T, W)$ in $\mathcal{G}_{\underline{Z_{\text{CFD}}}}$ (Rule 2 in Theorem 5)

$P(t' \mid do(z_{\text{CFD}}), w) = P(t' \mid w)$,

  since $(T \perp\!\!\!\perp Z_{\text{CFD}} \mid W)$ in $\mathcal{G}_{\overline{Z_{\text{CFD}}(W)}}$ (Rule 3 in Theorem 5)

$$P(y \mid do(t), z_{\text{CFD}}, w) = \sum_{t'} P(y \mid t', z_{\text{CFD}}, w)P(t' \mid w) \tag{12}$$

The second part: $P(w \mid do(t), z_{\text{CFD}}) = P(w, z_{\text{CFD}} \mid do(t))/P(z_{\text{CFD}} \mid do(t))$

$P(w, z_{\text{CFD}} \mid do(t)) = P(z_{\text{CFD}} \mid t, w)P(w)$

$$P(w \mid do(t), \mathbf{z}_{\text{FD}}) = \frac{P(z_{\text{CFD}} \mid t, w)P(w)}{P(z_{\text{CFD}} \mid do(t))} \tag{13}$$

$$\text{Thus, } P(y \mid z_{\text{CFD}}, do(t)) = \sum_{w,t'} P(y \mid t', z_{\text{CFD}}, w)P(t' \mid w)\frac{P(z_{\text{CFD}} \mid t, w)P(w)}{P(z_{\text{CFD}} \mid do(t))} \tag{14}$$

We take Eq. 14 into Eq. 10 and get,

$$P(y \mid do(t)) = \sum_{z_{\text{CFD}}} P(z_{\text{CFD}} \mid do(t)) \sum_{w,t'} P(y \mid t', z_{\text{CFD}}, w)P(t' \mid w)\frac{P(z_{\text{CFD}} \mid t, w)P(w)}{P(z_{\text{CFD}} \mid do(t))}$$

$$= \sum_{z_{\text{CFD}},w,t'} P(z_{\text{CFD}} \mid do(t))P(y \mid t', z_{\text{CFD}}, w)P(t' \mid w)\frac{P(z_{\text{CFD}} \mid t, w)P(w)}{P(z_{\text{CFD}} \mid do(t))} \tag{15}$$

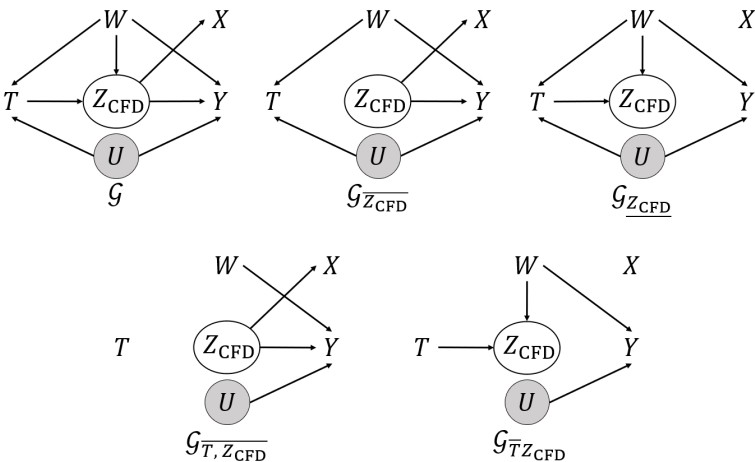

Figure 5: Subgraphs of $\mathcal{G}$ used in the derivation of causal effects.

Finally, we get,

$$P(y \mid do(t)) = \sum_{z_{\text{CFD}}, w, t'} P(y \mid t', z_{\text{CFD}}, w) P(t' \mid w) P(z_{\text{CFD}} \mid t, w) P(w) \tag{16}$$

where $t'$ is a distinct realisation of treatment. $\qquad\square$

### B.2 Proof of Theorem 4

Our proof is based on the proof of Theorem 1 in (Khemakhem et al., 2020).

*Proof.* Suppose we have two sets of parameters $(\text{f}, \text{S}, \boldsymbol{\lambda})$ and $(\widetilde{\text{f}}, \widetilde{\text{S}}, \widetilde{\boldsymbol{\lambda}})$ such that $p_{\boldsymbol{\theta}}(X, Z_{\text{CFD}} \mid T, W) = p_{\widetilde{\boldsymbol{\theta}}}(X, Z_{\text{CFD}} \mid T, W)$. Then:

$$\int_{\mathcal{Z}_{\text{CFD}}} p_{\text{S}, \boldsymbol{\lambda}}(Z_{\text{CFD}} \mid T, W) p_{\text{f}}(X \mid Z_{\text{CFD}}) \text{d}Z_{\text{CFD}}$$

$$= \int_{\mathcal{Z}_{\text{CFD}}} p_{\widetilde{\text{S}}, \widetilde{\boldsymbol{\lambda}}}(Z_{\text{CFD}} \mid T, W) p_{\widetilde{\text{f}}}(X \mid Z_{\text{CFD}}) \text{d}Z_{\text{CFD}}$$

$$\implies \int_{\mathcal{Z}_{\text{CFD}}} p_{\text{S}, \boldsymbol{\lambda}}(Z_{\text{CFD}} \mid T, W) p_{\boldsymbol{\varepsilon}}(X - \text{f}(Z_{\text{CFD}})) \text{d}Z_{\text{CFD}}$$

$$= \int_{\mathcal{Z}_{\text{CFD}}} p_{\widetilde{\text{S}}, \widetilde{\boldsymbol{\lambda}}}(Z_{\text{CFD}} \mid T, W) p_{\boldsymbol{\varepsilon}}(X - \widetilde{\text{f}}(Z_{\text{CFD}})) \text{d}Z_{\text{CFD}}$$

$$\implies \int_{\mathcal{X}} p_{\text{S}, \boldsymbol{\lambda}}(\text{f}^{-1}(\bar{X}) \mid T, W) \text{vol} J_{\text{f}^{-1}}(\bar{X}) p_{\boldsymbol{\varepsilon}}(X - \bar{X}) \text{d}\bar{X}$$

$$= \int_{\mathcal{X}} p_{\widetilde{\text{S}}, \widetilde{\boldsymbol{\lambda}}}(\widetilde{\text{f}}^{-1}(\bar{X}) \mid T, W) \text{vol} J_{\widetilde{\text{f}}^{-1}}(\bar{X}) p_{\boldsymbol{\varepsilon}}(X - \bar{X}) \text{d}\bar{X} \tag{17}$$

We denote the volume of a matrix vol $\boldsymbol{A}$, and when $\boldsymbol{A}$ is full column rank, vol $\boldsymbol{A} = \sqrt{\det \boldsymbol{A}^T \boldsymbol{A}}$. $J$ denotes the Jacobian, and we make the change of the variable $\bar{X} = \text{f}(Z_{\text{CFD}})$ on the left hand side, and $\bar{X} = \widetilde{\text{f}}(Z_{\text{CFD}})$ on the right hand side.

From Eq. 17, we have:

$$p_{S,\boldsymbol{\lambda}}(f^{-1}(\bar{X}) \mid T, W)\mathrm{vol}J_{f^{-1}}(\bar{X}) = p_{\widetilde{S},\widetilde{\boldsymbol{\lambda}}}(\widetilde{f}^{-1}(\bar{X}) \mid T, W)\mathrm{vol}J_{\widetilde{f}^{-1}}(\bar{X}) \tag{18}$$

By taking the logarithm on the both sides of Eq. 18 and replacing $p_{S,\boldsymbol{\lambda}}$ by its expression from Eq. 8, we get:

$$\log \mathrm{vol}J_{f^{-1}}(X) + \sum_{i=1}^{n}(\log Q_i(f_i^{-1}(X)) - \log Z_i(T, W) + \sum_{j=1}^{k} S_{i,j}(f_i^{-1}(X))\lambda_{i,j}(T, W)) =$$

$$\log \mathrm{vol}J_{\widetilde{f}^{-1}}(X) + \sum_{i=1}^{n}(\log \widetilde{Q}_i(\widetilde{f}_i^{-1}(X)) - \log \widetilde{Z}_i(T, W) + \sum_{j=1}^{k} \widetilde{S}_{i,j}(\widetilde{f}_i^{-1}(X))\widetilde{\lambda}_{i,j}(T, W)) \tag{19}$$

Let $(T, W)_0, ..., (T, W)_{nk}$ be the points provided by Theorem 4 (4), and define $\bar{\boldsymbol{\lambda}}(T, W) = \boldsymbol{\lambda}(T, W) - \boldsymbol{\lambda}(T_0, W_0)$. We plug each of those $(T, W)_l$ in Eq. 19 to obtain $nk + 1$ such equations. We subtract the first equation for $(T, W)_0$ from the remaining $nk$ equations to get for $l = 1, ..., nk$:

$$\langle S(f^{-1}(X)), \bar{\boldsymbol{\lambda}}(T_l, W_l)\rangle + \sum_i \log \frac{Z_i(T_0, W_0)}{Z_i(T_l, W_l)}$$

$$= \langle \widetilde{S}(\widetilde{f}^{-1}(X)), \bar{\widetilde{\boldsymbol{\lambda}}}(T_l, W_l)\rangle + \sum_i \log \frac{\widetilde{Z}_i(T_0, W_0)}{\widetilde{Z}_i(T_l, W_l)} \tag{20}$$

Let $\boldsymbol{L}$ be the matrix defined in Theorem 4 (4), and $\widetilde{\boldsymbol{L}}$ similarly defined for $\widetilde{\boldsymbol{\lambda}}$ ($\widetilde{\boldsymbol{L}}$ is not necessarily invertible). Define $b_l = \sum_i \log \frac{\widetilde{Z}_i(T_0, W_0)Z_i(T_l, W_l)}{Z_i(T_0, W_0)\widetilde{Z}_i(T_l, W_l)}$ and b the vector of all $b_l$ for $l = 1, ..., nk$.

Then, Eq. 20 can be rewritten as:

$$\boldsymbol{L}^T S(f^{-1}(X)) = \widetilde{\boldsymbol{L}}^T \widetilde{S}(\widetilde{f}^{-1}(X)) + b \tag{21}$$

We multiply both sides of Eq. 21 by the transpose of the inverse of $\boldsymbol{L}^T$ from the left to get:

$$S(f^{-1}(X)) = \boldsymbol{A}\widetilde{S}(\widetilde{f}^{-1}(X)) + c, \tag{22}$$

where $\boldsymbol{A} = \boldsymbol{L}^{-T}\widetilde{\boldsymbol{L}}$ and c $= \boldsymbol{L}^{-T}b$.

By definition of S and according to Theorem 4 (3), its Jacobian exists and is an $nk \times n$ matrix of rank $n$. This implies that the Jacobian of $\widetilde{S} \circ f^{-1}$ exists and is of rank $n$ and so is $\boldsymbol{A}$. We have two cases: (1) If $k = 1$, $\boldsymbol{A}$ is invertible since $\boldsymbol{A}$ is $n \times n$ matrix of rank $n$; (2) If $k >= 2$, $\boldsymbol{A}$ is also invertible. We have the following proof for (2):

Define $\bar{X} = f^{-1}(X)$ and $S_i(\bar{X}_i) = (S_{i,1}(\bar{X}_i), ..., S_{i,k}(\bar{X}_i))$. For each $i \in [1, ..., n]$ there exist $k$ points $\bar{X}_i^1, ..., \bar{X}_i^k$ such that $(S_i'(\bar{X}_i^1), ..., S_i'(\bar{X}_i^k))$ are linearly independent.

Firstly, we proof the above statement. Suppose that for any choice of such $k$ points, the family $(S_i'(\bar{X}_i^1), ..., S_i'(\bar{X}_i^k))$ is never linearly independent. That means that $S_i'(\mathbb{R})$ is included in a subspace of $\mathbb{R}^k$ of dimension at most $k - 1$. Let $\boldsymbol{h}$ a non zero vector that is orthogonal to $S_i'(\mathbb{R})$. Then for all $X \in \mathbb{R}$, we have $\langle S_i'(\mathbb{R}), \boldsymbol{h}\rangle = 0$. By integrating we find that $\langle S_i(\mathbb{R}), \boldsymbol{h}\rangle = \mathrm{const}$. Since this is true for all $X \in \mathbb{R}$ and for a $\boldsymbol{h} \neq 0$, we conclude that the distribution is not strongly exponential, which contradicts our hypothesis.

Secondly, we prove $\boldsymbol{A}$ is invertible. Collect those points into $k$ vectors $(\bar{X}^1, ..., \bar{X}^k)$, and concatenate the $k$ Jacobians $J_S(\bar{X}^l)$ evaluated at each of those vectors horizontally into the matrix $\boldsymbol{Q} = (J_S(\bar{X}^1), ..., J_S(\bar{X}^k))$ (and similarly define $\widetilde{\boldsymbol{Q}}$ as the concatenation of the Jacobians of $\widetilde{S}(\widetilde{f}^{-1} \circ f(\bar{X}))$ evaluated at those points). Then the matrix $\boldsymbol{Q}$ is invertible. By differentiating Eq. 22 for each $X^l$, we have:

$$\boldsymbol{Q} = \boldsymbol{A}\widetilde{\boldsymbol{Q}} \tag{23}$$

The invertibility of $\boldsymbol{Q}$ implies the invertibility of $\boldsymbol{A}$ and $\widetilde{\boldsymbol{Q}}$, which completes the proof. $\square$

### B.3 The Validity of the Learned CFD Adjustment Variable

Let us consider the linear causal relationships where $T$ affects $Y$ through the (conditional) front-door adjustment variable $Z_{\text{CFD}}$, as represented as follows:

$$Z_{\text{CFD}} = \alpha T + \epsilon_{Z_{\text{CFD}}} \tag{24}$$

$$Y = \beta Z_{\text{CFD}} + \epsilon_Y \tag{25}$$

where $\epsilon_{Z_{\text{CFD}}}$ and $\epsilon_Y$ are the error terms, assumed to be mean-zero, independent random errors. The causal effect of $T$ on $Y$ is commonly computed as the product of the path coefficients, i.e., ATE $= \alpha\beta$.

Following the discussion in Section 4.3, our proposed method CFDiVAE guarantees the model identifiability, which means that the learned latent CFD adjustment variable $\widetilde{Z_{\text{CFD}}} = \widetilde{\text{f}}^{-1}(X)$ is equal to the ground truth CFD adjustment variable $Z_{\text{CFD}} = \text{f}^{-1}(X)$, up to a linear invertible transformation (through an invertible matrix $\boldsymbol{A}$) and point-wise nonlinearities (in the form of S and $\widetilde{\text{S}}$), that is,

$$Z_{\text{CFD}} = \boldsymbol{A}\widetilde{Z_{\text{CFD}}} \tag{26}$$

Substituting the above expression for $\widetilde{Z_{\text{CFD}}}$ into Eq. 24, we get,

$$\boldsymbol{A}\widetilde{Z_{\text{CFD}}} = \alpha T + \epsilon_{Z_{\text{CFD}}} \tag{27}$$

Multiplying both sides of the Eq. 27 by $\boldsymbol{A}^{-1}$, we get,

$$\boldsymbol{A}^{-1}\boldsymbol{A}\widetilde{Z_{\text{CFD}}} = \widetilde{Z_{\text{CFD}}} = \alpha\boldsymbol{A}^{-1}T + \boldsymbol{A}^{-1}\epsilon_{Z_{\text{CFD}}} \tag{28}$$

Substituting the expression for $\widetilde{Z_{\text{CFD}}}$ shown in Eq. 26 into Eq. 25, we get,

$$Y = \beta\boldsymbol{A}\widetilde{Z_{\text{CFD}}} + \epsilon_Y \tag{29}$$

Eq. 28 shows that the causal effect of $T$ on $\widetilde{Z_{\text{CFD}}}$ is $\alpha\boldsymbol{A}^{-1}$, and Eq. 29 shows that the causal effct of $\widetilde{Z_{\text{CFD}}}$ on $Y$ is $\beta\boldsymbol{A}$, so the causal effect of $T$ on $Y$ via $\widetilde{Z_{\text{CFD}}}$ is the product of the two coefficients, i.e., $\alpha\boldsymbol{A}^{-1}\beta\boldsymbol{A} = \alpha\beta\boldsymbol{A}^{-1}\boldsymbol{A} = \alpha\beta$. This means that using the learned (transformed) CFD adjustment variable will give us the same ATE as using the groundtruth CFD adjustment variable.

## C Description of ATE Estimation

For discrete data, we show the solution in Algorithm 1. We see that the solution is aligned with Eq. 3.

For the continues data with linear model, we assume the data generation as followed,

$$Z_{\text{CFD}} = c_{Z_{\text{CFD}}} + \beta_{T,Z_{\text{CFD}}}T + \beta_{W,Z_{\text{CFD}}}W + e_{Z_{\text{CFD}}}; \tag{30}$$

$$Y = c_Y + \beta_{Y,Z_{\text{CFD}}}Z_{\text{CFD}} + \beta_{W,Y}W + \beta_{U,Y}U + e_Y, \tag{31}$$

where $c$ denotes intercept, $e$ denotes error, and $Z_{\text{CFD}}$ is the learned representation by our method.

The ATE of $T$ on $Y$ is the product of coefficients $\beta_{T,Z_{\text{CFD}}}$ and $\beta_{Y,Z_{\text{CFD}}}$. The coefficients are obtained with the following process (Tchetgen & Shpitser, 2012; Barr, 2018):

**Step 1**: $Z_{\text{CFD}}$ is regressed on $T$ and $W$. This gives us the coefficient $\beta_{T,Z_{\text{CFD}}}$ and $\mathbb{E}[Z_{\text{CFD}} \mid T, W]$.

**Step 2**: Using $\mathbb{E}[Z_{\text{CFD}} \mid T, W]$, we estimate the noise $e_{Z_{\text{CFD}}}$ as $Z_{\text{CFD}} - \mathbb{E}[Z_{\text{CFD}} \mid T, W]$. Regress $e_{Z_{\text{CFD}}}$ on $Y$. This gives us the coefficient $\beta_{Y,Z_{\text{CFD}}}$. Noise $e_{Z_{\text{CFD}}}$ is only introduced at $Z_{\text{CFD}}$, and is independent of the unobserved confounding variable $U$.

Although the solution for continuous data with the linear model does not mirror Eq. 3, it is still consistent with the (conditional) front-door adjustment shown in Theorem 3. As we know, the

---

**Algorithm 1** CFD Adjustment for ATE Estimation with Discrete Data

---

1: **Input:** $T$, $Y$, $W$, and $Z_{\text{CFD}}$
2: **Output:** ATE of $T$ on $Y$
3: Initialise $\mathbb{E}[Y|do(T=0)]$ and $\mathbb{E}[Y|do(T=1)]$
4: **for** each value of $T$ **do**
5:     Estimate the marginal probability $P(t)$
6: **end for**
7: **for** each combination of $T$, $W$, and $Z_{\text{CFD}}$ **do**
8:     Estimate the conditional probability $P(z_{\text{CFD}}|t,w)$
9: **end for**
10: **for** each combination of $T$, $W$, $Y$, and $Z_{\text{CFD}}$ **do**
11:     Estimate the conditional probability $P(y|t,w,z_{\text{CFD}})$
12: **end for**
13: **for** each value of $Y$, $W$, and $Z_{\text{CFD}}$ **do**
14:     **for** each value of $T$ **do**
15:         $\mathbb{E}[Y|do(T=0)] \mathrel{+}= yP(y|t,z_{\text{CFD}},w)P(z_{\text{CFD}}|T=0,w)P(t|w)P(w)$
16:         $\mathbb{E}[Y|do(T=1)] \mathrel{+}= yP(y|t,z_{\text{CFD}},w)P(z_{\text{CFD}}|T=1,w)P(t|w)P(w)$
17:     **end for**
18: **end for**
19: Compute the ATE as $\mathbb{E}[Y|do(T=1)] - \mathbb{E}[Y|do(T=0)]$
20: **return** ATE

---

front-door adjustment consists of two back-door adjustments. In our case with continuous variables, these two back-door adjustments are implemented by regression adjustments, as shown in Eq. 30 and Eq. 31. Specifically, for the first backdoor adjustment, we use regression adjustment to get the causal effect of $T$ on $Z_{\text{CFD}}$ (i.e., $\beta_{T,Z_{\text{CFD}}}$) adjusting on $W$, as shown in Eq. 30 and Step 1 of the process; for the second backdoor adjustment, we also use regression adjustment as shown in Eq. 31 to obtain the causal effect of $Z_{\text{CFD}}$ on $Y$ adjusting on $W$ and $U$. However, for this regression adjustment, since $U$ is unobserved, we follow the approach in (Tchetgen & Shpitser, 2012; Barr, 2018) to firstly obtain the error $e_{Z_{\text{CFD}}}$, which is independent of $U$, and regress $e_{Z_{\text{CFD}}}$ on $Y$ instead. In this way, we can eliminate the confounding bias caused by $U$. This is shown in Step 2 of the process.

## D EXPERIMENT

### D.1 DESCRIPTION OF THE COMPARISON METHODS

**LinearDRL** Chernozhukov et al. (2018): A double machine learning estimator with a low-dimensional linear regression as the final stage.

**CausalForest** Wager & Athey (2018): A causal forest estimator combined with the double machine learning technique for conditional average treatment effect estimation.

**ForestDRL** Athey et al. (2019): A generalised random forest and orthogonal random forest based estimator that uses doubly-robust correction techniques to account for covariates shift (or selection bias) between the treatment.

**XLearner** Künzel et al. (2019): A meta-learning algorithm that utilises supervised learning methods (e.g., Random Forests and Bayesian Regression) for the analysis of conditional average treatment effects.

**KernelDML** Nie & Wager (2021): A specialised version of the double machine learning estimator that uses random fourier features and kernel ridge regression for the analysis of conditional average treatment effects.

**CEVAE** Louizos et al. (2017): A deep learning based method that leverages latent variable modelling, specifically Variational AutoEncoder, to estimate causal effect from observational data, even in the presence of latent confounders.

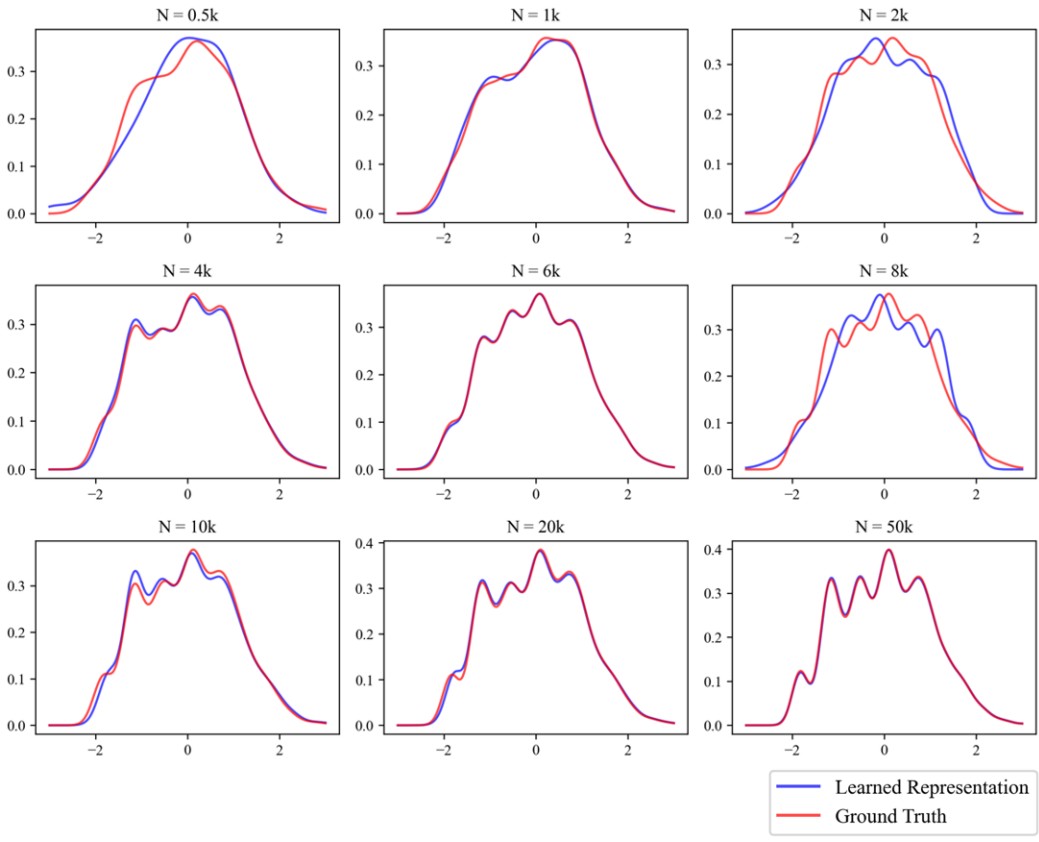

Figure 6: Probability Density Functions of the ground truth CFD adjustment variable and the learned representation, where the horizontal axis represents the value and the vertical axis represents the density.

**TEDVAE** Zhang et al. (2021): A deep learning based method that learns the disentangled representations of confounding, instrumental, and risk factors using VAE for accurate treatment effect estimation.

### D.2    MORE RESULTS OF THE EXPERIMENTS IN SECTION 6.2

In this section, we compare the probability distribution of the learned representation of the CFD adjustment variable with the distribution of the ground truth CFD adjustment variable under different sample sizes. As shown in Fig. 6, the distribution of the learned representation is close to the ground truth distribution, which indicates that the proposed method CFDiVAE can learn the accurate representation of the CFD adjustment variable from its proxy.

### D.3    ANALYSIS OF MODEL IDENTIFIABILITY

Our proposed method CFDiVAE takes $T$ and $W$ as additional observed variables to approximate the prior $p(Z_{\mathrm{CFD}} \mid T, W)$. In this section, we apply two partially identifiable VAE models, i.e., $T$-CFDiVAE and $W$-CFDiVAE, and the original VAE as comparison methods. $T$-CFDiVAE is partially identifiable VAE model that takes $T$ as the additional observed variable to approximate $p(Z_{\mathrm{CFD}} \mid T)$; $W$-CFDiVAE is partially identifiable VAE model that takes $W$ as the additional observed variable to approximate $p(Z_{\mathrm{CFD}} \mid W)$; the original VAE does not take any additional observed variable to approximate $p(Z_{\mathrm{CFD}})$. The ELBOs for these methods are defined as:

Table 4: Results of model identifiability analysis.

| | 0.5k | 1k | 2k | 4k | 6k | 8k | 10k | 20k |
|---|---|---|---|---|---|---|---|---|
| $\mathcal{M}_{\text{VAE}}$ (Eq. 34) | 88.93 ± 14.67 | 58.41 ± 22.46 | 77.71 ± 9.17 | 79.89 ± 7.00 | 32.92 ± 31.20 | 10.49 ± 7.66 | 7.07 ± 5.72 | 4.55 ± 9.19 |
| $\mathcal{M}_{T\text{-CFDiVAE}}$ (Eq. 32) | 70.09 ± 20.60 | 77.48 ± 12.65 | 81.48 ± 8.19 | 15.94 ± 22.84 | 10.25 ± 22.99 | 4.60 ± 6.11 | 3.89 ± 4.56 | 1.77 ± 1.41 |
| $\mathcal{M}_{W\text{-CFDiVAE}}$ (Eq. 33) | 68.89 ± 18.89 | 85.83 ± 4.64 | 29.13 ± 21.00 | 5.49 ± 3.83 | 6.49 ± 17.82 | 3.14 ± 2.04 | 3.14 ± 2.35 | 1.69 ± 1.55 |
| $\mathcal{M}_{\text{CFDiVAE}}$ (Eq. 4) | 86.29 ± 6.21 | 39.72 ± 31.47 | 8.87 ± 10.68 | 4.57 ± 3.03 | 2.58 ± 1.96 | 2.32 ± 1.47 | 2.97 ± 2.09 | 1.57 ± 1.32 |

$$\mathcal{M}_{T\text{-CFDiVAE}} = \mathbb{E}_q[\log p(X \mid Z_{\text{CFD}})] - D_{\text{KL}}[q(Z_{\text{CFD}} \mid T, X) \,\|\, p(Z_{\text{CFD}} \mid T)] \quad (32)$$

$$\mathcal{M}_{W\text{-CFDiVAE}} = \mathbb{E}_q[\log p(X \mid Z_{\text{CFD}})] - D_{\text{KL}}[q(Z_{\text{CFD}} \mid W, X) \,\|\, p(Z_{\text{CFD}} \mid W)] \quad (33)$$

$$\mathcal{M}_{\text{VAE}} = \mathbb{E}_q[\log p(X \mid Z_{\text{CFD}})] - D_{\text{KL}}[q(Z_{\text{CFD}} \mid X) \,\|\, p(Z_{\text{CFD}})] \quad (34)$$

The results are shown in Table 4. We see that CFDiVAE achieves the best performance since it uses all additional observed variables. The performance of $T$-CFDiVAE is slightly lower than the performance of $W$-CFDiVAE since $W$ has more additional information than $T$ (the dimension of $W$ is generally higher than the dimension of $T$). The original VAE which does not use any additional observed variable achieves the worst performance.

### D.4   MORE RESULTS OF THE EXPERIMENTS IN SECTION 6.5

In this section, we evaluate the performance of CFDiVAE and the comparison methods when the dimension for the representation does not match the dimension of the ground truth CFD adjustment variable. Table 5a shows the results for $D_{\text{R}} = 2$, Table 5b shows the results for $D_{\text{R}} = 4$, and Table 5c shows the results for $D_{\text{R}} = 8$. We note that CFDiVAE achieves its best performance when $D_{\text{L}} = D_{\text{X}}$ in most sample sizes. Hence, in a more general case, we can safely set $D_{\text{L}} = D_{\text{X}}$ to get an acceptable causal effect estimation.

### D.5   CAUSAL GRAPHS INVOLVED IN SECTION 6.6

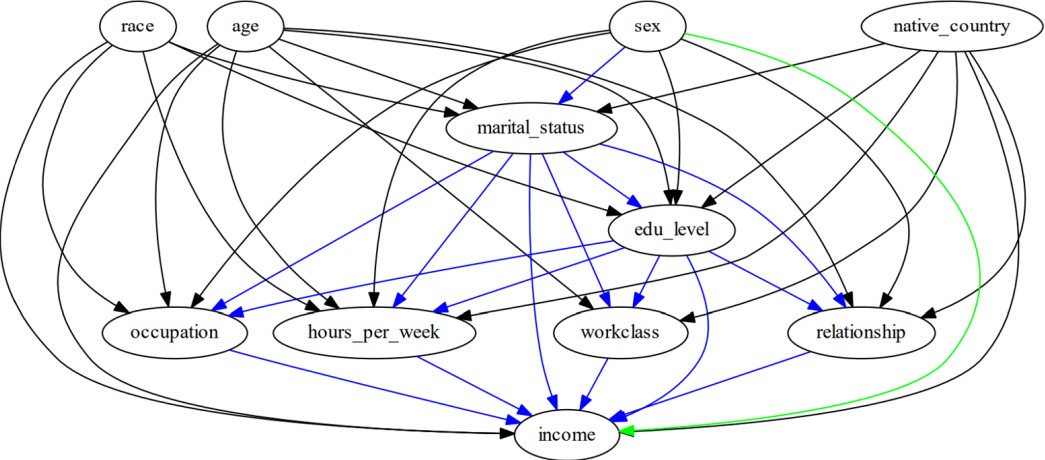

Figure 7: The causal network for the Adult dataset: the green path represents the direct path, and the blue paths represent the indirect paths passing through marital_status (Zhang et al., 2017a).

Table 5: The estimation bias (%) of CFDiVAE and comparison methods under different $N$ values. CFDiVAE-$D_\mathrm{L}$-$D_\mathrm{R}$-$D_\mathrm{X}$ denotes apply CFDiVAE to a specified setting, where $D_\mathrm{L}$ represents the dimension of the learned representation, $D_\mathrm{R}$ represents the dimension of the ground truth CFD adjustment variable, and $D_\mathrm{X}$ represents the dimension of the proxy variable.

(a) Estimation bias (%) when $D_\mathrm{R} = 2$.

| | 0.5k | 1k | 2k | 4k | 6k | 8k | 10k | 20k |
|---|---|---|---|---|---|---|---|---|
| LinearDRL | 27.05 ± 7.56 | 24.58 ± 6.39 | 26.13 ± 4.19 | 24.53 ± 3.01 | 25.01 ± 2.90 | 24.92 ± 1.61 | 25.40 ± 1.56 | 25.42 ± 1.12 |
| CausalForest | 28.26 ± 8.21 | 24.51 ± 6.76 | 26.01 ± 4.48 | 24.56 ± 3.11 | 25.09 ± 3.10 | 24.98 ± 1.58 | 25.40 ± 1.67 | 25.46 ± 1.13 |
| ForestDRL | 26.91 ± 7.95 | 24.51 ± 6.38 | 26.20 ± 3.99 | 24.54 ± 3.11 | 24.95 ± 2.87 | 24.96 ± 1.60 | 25.38 ± 1.59 | 25.43 ± 1.13 |
| XLearn | 27.15 ± 7.50 | 24.62 ± 6.29 | 26.04 ± 3.94 | 24.57 ± 3.09 | 25.02 ± 2.90 | 24.94 ± 1.59 | 25.37 ± 1.56 | 25.40 ± 1.11 |
| KernelDML | 24.34 ± 7.84 | 22.01 ± 6.29 | 24.23 ± 4.09 | 22.94 ± 3.05 | 23.57 ± 2.81 | 23.57 ± 1.47 | 24.06 ± 1.58 | 24.22 ± 1.08 |
| CEVAE | 102.05 ± 3.22 | 104.47 ± 9.79 | 104.04 ± 22.15 | 41.27 ± 7.16 | 39.88 ± 8.89 | 32.34 ± 10.46 | 23.62 ± 11.21 | 34.20 ± 6.46 |
| TEDVAE | 93.05 ± 14.19 | 69.77 ± 21.04 | 24.61 ± 11.13 | 29.14 ± 2.81 | 26.55 ± 2.99 | 25.98 ± 1.50 | 26.22 ± 1.55 | 26.03 ± 1.30 |
| CFDiVAE-1-2-4 | 82.31 ± 8.83 | 11.99 ± 5.98 | 10.70 ± 17.07 | 9.52 ± 3.08 | 9.54 ± 2.34 | 9.86 ± 2.54 | 10.35 ± 4.25 | 9.88 ± 1.36 |
| CFDiVAE-2-2-4 | 78.16 ± 4.99 | 12.85 ± 10.96 | 6.90 ± 5.88 | 8.83 ± 6.02 | 5.94 ± 4.22 | 5.46 ± 3.62 | 5.37 ± 6.82 | 4.16 ± 8.90 |
| CFDiVAE-4-2-4 | 76.69 ± 4.44 | 23.21 ± 15.38 | 12.31 ± 19.33 | 7.98 ± 6.98 | 5.73 ± 5.45 | 7.45 ± 3.36 | 3.90 ± 4.87 | 1.77 ± 1.08 |

(b) Estimation bias (%) when $D_\mathrm{R} = 4$.

| | 0.5k | 1k | 2k | 4k | 6k | 8k | 10k | 20k |
|---|---|---|---|---|---|---|---|---|
| LinearDRL | 32.82 ± 13.77 | 30.76 ± 10.62 | 32.91 ± 7.69 | 32.38 ± 4.51 | 32.38 ± 3.69 | 33.04 ± 3.33 | 32.13 ± 3.30 | 31.97 ± 1.98 |
| CausalForest | 32.40 ± 13.57 | 31.34 ± 11.30 | 33.41 ± 7.79 | 32.88 ± 4.25 | 32.49 ± 3.94 | 33.18 ± 3.16 | 32.11 ± 3.36 | 31.94 ± 1.93 |
| ForestDRL | 32.42 ± 13.57 | 31.21 ± 10.75 | 32.78 ± 8.07 | 32.28 ± 4.31 | 32.43 ± 3.76 | 32.92 ± 3.20 | 32.13 ± 3.47 | 32.01 ± 1.97 |
| XLearn | 32.88 ± 12.93 | 31.14 ± 10.54 | 32.81 ± 7.65 | 32.36 ± 4.37 | 32.36 ± 3.69 | 32.98 ± 3.33 | 32.15 ± 3.35 | 32.00 ± 1.99 |
| KernelDML | 28.12 ± 14.57 | 27.83 ± 10.53 | 29.48 ± 7.97 | 30.47 ± 4.24 | 30.32 ± 3.72 | 31.19 ± 3.33 | 30.48 ± 3.17 | 30.60 ± 1.98 |
| CEVAE | 102.10 ± 2.45 | 102.03 ± 13.14 | 123.90 ± 39.63 | 34.56 ± 28.24 | 65.52 ± 16.65 | 50.36 ± 15.35 | 35.41 ± 17.62 | 42.41 ± 10.94 |
| TEDVAE | 99.99 ± 16.35 | 75.63 ± 40.63 | 30.72 ± 21.25 | 44.30 ± 4.87 | 35.52 ± 3.88 | 35.33 ± 3.24 | 33.82 ± 3.44 | 33.12 ± 2.20 |
| CFDiVAE-1-4-8 | 79.94 ± 8.98 | 22.12 ± 18.63 | 12.09 ± 4.62 | 13.73 ± 3.58 | 14.24 ± 3.43 | 15.07 ± 2.86 | 14.33 ± 2.64 | 14.83 ± 1.74 |
| CFDiVAE-2-4-8 | 74.31 ± 6.90 | 16.38 ± 8.40 | 9.49 ± 5.02 | 11.54 ± 3.75 | 9.84 ± 3.15 | 8.19 ± 4.86 | 8.43 ± 6.85 | 6.10 ± 1.83 |
| CFDiVAE-4-4-8 | 73.16 ± 5.70 | 19.04 ± 11.12 | 12.89 ± 16.47 | 9.90 ± 5.15 | 8.74 ± 5.69 | 6.78 ± 3.92 | 4.50 ± 2.70 | 4.45 ± 1.75 |
| CFDiVAE-8-4-8 | 69.66 ± 8.20 | 31.97 ± 12.11 | 38.96 ± 12.17 | 11.73 ± 4.09 | 8.05 ± 6.67 | 6.07 ± 3.75 | 4.42 ± 2.46 | 4.44 ± 2.00 |

(c) Estimation bias (%) when $D_\mathrm{R} = 8$.

| | 0.5k | 1k | 2k | 4k | 6k | 8k | 10k | 20k |
|---|---|---|---|---|---|---|---|---|
| LinearDRL | 54.02 ± 30.87 | 48.22 ± 16.82 | 47.84 ± 13.83 | 48.01 ± 9.29 | 46.89 ± 7.52 | 47.17 ± 4.97 | 48.66 ± 6.26 | 47.56 ± 3.31 |
| CausalForest | 51.85 ± 31.93 | 49.64 ± 17.88 | 47.48 ± 13.26 | 47.06 ± 9.55 | 47.65 ± 7.71 | 47.68 ± 5.13 | 48.85 ± 6.41 | 47.47 ± 3.54 |
| ForestDRL | 53.42 ± 31.29 | 47.92 ± 17.51 | 47.76 ± 13.55 | 48.46 ± 9.34 | 46.74 ± 7.53 | 47.20 ± 4.83 | 48.60 ± 6.19 | 47.59 ± 3.34 |
| XLearn | 53.26 ± 30.78 | 48.08 ± 17.18 | 47.74 ± 13.99 | 48.06 ± 9.23 | 46.88 ± 7.56 | 47.15 ± 4.88 | 48.61 ± 6.23 | 47.54 ± 3.28 |
| KernelDML | 46.03 ± 31.34 | 43.16 ± 17.91 | 42.93 ± 13.83 | 44.79 ± 9.85 | 43.50 ± 7.86 | 44.59 ± 4.81 | 46.30 ± 6.26 | 45.65 ± 3.21 |
| CEVAE | 101.71 ± 2.29 | 107.12 ± 13.68 | 122.72 ± 46.09 | 106.34 ± 69.06 | 94.41 ± 29.52 | 106.92 ± 26.61 | 92.49 ± 36.07 | 33.48 ± 17.93 |
| TEDVAE | 98.58 ± 22.10 | 74.62 ± 55.07 | 60.77 ± 37.72 | 80.72 ± 11.28 | 59.95 ± 7.74 | 51.93 ± 5.16 | 52.78 ± 6.30 | 49.85 ± 3.43 |
| CFDiVAE-1-8-16 | 75.38 ± 12.60 | 27.92 ± 22.61 | 15.86 ± 6.77 | 14.46 ± 17.05 | 15.18 ± 4.85 | 16.62 ± 3.73 | 16.64 ± 3.34 | 17.65 ± 2.29 |
| CFDiVAE-4-8-16 | 72.33 ± 11.25 | 19.45 ± 13.21 | 12.89 ± 10.46 | 11.71 ± 12.01 | 12.28 ± 5.88 | 10.36 ± 5.37 | 9.26 ± 5.88 | 7.47 ± 3.04 |
| CFDiVAE-8-8-16 | 63.95 ± 11.41 | 27.47 ± 13.96 | 29.00 ± 26.66 | 11.01 ± 11.10 | 10.00 ± 7.61 | 9.00 ± 6.27 | 6.69 ± 4.29 | 7.42 ± 3.14 |
| CFDiVAE-16-8-16 | 56.84 ± 11.64 | 22.67 ± 12.68 | 17.15 ± 19.49 | 10.60 ± 9.62 | 7.86 ± 6.56 | 8.28 ± 5.75 | 6.23 ± 4.15 | 5.30 ± 2.99 |

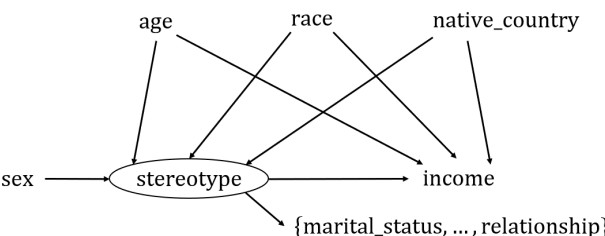

Figure 8: Simplified DAG for Adult dataset.

# E   REPRODUCIBILITY

Table 6: Details of the parameter settings in CFDiVAE.

| Parameter | Value | Parameter | Value | Parameter | Value |
|---|---|---|---|---|---|
| Reps | 30 | Num_Layers | 3 | wd | 1e-4 |
| Epoch | 30 | lr | 1e-3 | | |
| Batch_Size | 256 | lrd | 0.01 | | |

In this section, we provide more details of the experimental setting and configuration for reproducibility purposes. CFDiVAE is implemented in Python (Van Rossum & Drake Jr, 1995) libraries PyTorch (Paszke et al., 2019) and Pyro (Bingham et al., 2019). The code for data generation is written in R (R Core Team, 2021). We provide the parameter settings of CFDiVAE in Table 6. The descriptions of the major parameters are provided below:

- Reps: the number of replications each set of experiments runs.
- Epoch: one Epoch is when an entire dataset is passed forward and backward through the neural network once.
- Batch_Size: the number of training examples present in a single batch.
- Num_Layers: the number of hidden layers.
- lr: the learning rate.
- lrd: the learning decay.
- wd: the weight decay.

