# OpenReview forum: "Causal Inference with Conditional Front-Door Adjustment and Identifiable Variational Autoencoder"
_ICLR.cc/2024/Conference — ICLR 2024 poster_

### Official Review · Reviewer_BvSP · 2023-10-26

**Soundness:** 2 fair
**Presentation:** 3 good
**Contribution:** 2 fair
**Rating:** 6
**Confidence:** 4

**Summary:**

POST REBUTTAL UPDATE
Overall the paper has clearly improved during the rebuttal phase. The assumptions and limitations of the method are much better discussed and the changes in the experiments have improved them. That said, I fully concur with reviewer s6a3 that discussion on sufficient statistics with which the theory holds should still be included. I will update my score to 6.
POST REBUTTAL UPDATE END

The paper considers front-door adjustment for causal effect estimation. In particular, the setup involves the following variables: T (treatment), W (observed confounders), U (unobserved confounders), Y the outcome, Z (front-door adjustment variable), and X proxy variable for Z. The article makes two main contributions: 1) The article proposes a conditional front-door adjustment, which differs from the standard front-door adjustment by having an additional edge from the observed confounders W to the front-door adjustment variable Z. The paper shows how in this situation the standard front-door adjustment must be modified to estimate the causal effect (Theorem 3). The proof is based on the assumption that the front-door adjustment variable Z is observed. 2) The second contribution is a conditional VAE method for the situation where the front-door adjustment variable Z is not observed, to estimate it from the proxies X (conditionally on variables W and T).

**Strengths:**

1) The first contribution, the conditional front-door adjustment, seems intuitive, interesting, and useful. To the best of my knowledge this is novel (though I haven’t read all the papers about front-door adjustment). The conditional front-door adjustment is presented clearly and the proof seems mathematically correct.

2) The VAE approach seems a reasonable attempt to estimate the front-door adjustment variable from the proxies.

3) The empirical results demonstrate that the proposed approach clearly outperforms methods that do not do any front-door adjustment at all. Sensitivity analysis for assuming an incorrect number of latent variables (smaller than correct) was a nice bonus.

**Weaknesses:**

1) There seems to be a gap in the theoretical part. The model identifiability analysis in Section 4.3 shows that the front-door adjustment variable Z obtained with VAE is identifiable up to a transformation (more or less the result from Khemakhem, 2020). But Theorem 3 is based on the assumption that the front-door adjustment variable Z is observed. It is not clear if Theorem 3 is still applicable if it is used for a transformed Z.

2) The empirical comparison includes as baselines only methods that do not use the proxies of the front-door variable at all, and consequently each of those has poor performance. It would be better to think how the proxies would be used if the new method did not exist, for example by using the proxies directly for front-door adjustment? The existing front-door adjustment methods (by Jeong, Wienobst) were not included in the comparison because they assumed that the DAG is known. However, also the present method assumes that the causal graph is the one presented in Figure 2 and all simulations assume this correct structure (except for a possible mismatch in the latent dimension).

3) Some small inconsistencies in the notation, at least: Equation (4): LHS has T and W but RHS does not. Equation (6): formulas for the mean and variance have j on the LHS but not on the RHS.

4) In the real-world example, the ATE estimate is 0.176, which is very similar to the previous estimate 0.175 (from Appendix C.5), which seems to imply that the new method does not provide too much novel insight in this case (but of course it is a good demonstration that it is consistent with previous estimates). In general, it would be nice to see some example where the heavy VAE machinery is really needed, which might require more complex, e.g., higher-dimensional, proxies, to make a convincing case.

Overall, I liked the first contribution, the conditional front-door adjustment, but I found the conditional VAE a bit confusing, possibly breaking the theory, and whose usefulness was not demonstrated very convincingly. I’m not sure but that paper might have been stronger without the VAE aspect altogether. At the very least the possible gap in the theory should be addressed. If it is not possible to fix the gap, then the theory (regarding the identifiability of the VAE) could be moved to the supplement and replaced by more convincing empirical analysis, and the text could be updated accordingly.

**Questions:**

1) Could you comment on the weakness number 1, please?

2) Section 6.2. Can you clarify what you mean by ground-truth density function for the representation? With simulated data, don’t you know the representation variable exactly for each data point, which would allow you to compare the estimated and the ground-truth values directly (e.g. using a scatter plot)?

3) What is the dimension of the proxy variable in the simulated experiments?

4) Can you clarify if you used the linear method in Section 5 to estimate the causal effect? How is this compatible with the conditional VAE model that parametrizes the model for the front-door adjustment variable with a neural network?

5) What would happen if you assumed a larger than correct dimension of the latent variable?

---

> ### Author Response · Authors · 2023-11-18
>
> Thanks for your review and insights.
>
> Regarding the first weakness, we think there might be some misunderstanding by the reviewer. We do not assume Z_CFD is observed. In fact, we consider Z_CFD to be unobserved and it is why CFDiVAE learns the representation of Z_CFD that satisfies the CFD criterion such that we can use the learned representation for CFD adjustment by following Theorem 3 (specifically Eqn. (3)). In other words, Theorem 3 states how to do CFD adjustment when a CFD adjustment variable is given, but this does not mean that our work assumes a CFD adjustment variable is known.
>
> For the second question, firstly we would like to clarify that in the paper, we used ground truth to refer to the ground truth adjustment "variable" or its distribution/density function (which is known since we generated the data, including the data of the adjustment variable), instead of the ground truth representation or its distribution. Then the purpose of the experiments in Section 6.2 is to compare the distribution of the learned representation of the adjustment variable with the distribution of the ground truth adjustment variable. In this way we can validate the correctness of the learned representation, if the representation is correctly learned, its distribution should be similar to that of the distribution of the ground truth adjustment variable.
>
> Regarding dimensionality, the proxy variables' dimensions we investigated are 4, 8, and 16. Following your comments, we conducted additional experiments on dimensionality analysis. The results show that CFDiVAE achieves its best performance in most sample sizes when $D_{\text{L}} = D_{\text{X}}$ (in this setting $D_{\text{L}} > D_{\text{R}}$, i.e., a larger setting than the correct dimension of the latent variable).
>
> About Section 5, CFDiVAE is for learning the representation of the CFD adjustment variable, then after Z_CFD has been learned, we can use it (as the adjustment variable) with any ATE estimation method and the choice of the ATE estimation method is based on the data generation (e.g., linear, or non-linear). For linear models, we propose a basic solution in the main paper. For nonlinear models, we are aware that semi-parametric methods could be applicable, but delving into these methods is outside the scope of our contributions.
>
> Following your suggestions, we addressed the inconsistencies in the notation and added two comparison methods based on the standard front door adjustment.
>
> Please find the latest version in the Rebuttal Revision.

---

> > ### Comment · Reviewer_BvSP · 2023-11-20
> > **Thanks for the comments**
> >
> > Thanks for your replies.
> >
> > I'm still missing this related to the first weakness:
> > Thm 4 guarantees the identifiability of the representation of the conditional front-door variable _up to a certain transformation_, right? Thm 3 guarantees that, given the CFD variable, you can identify the causal effect. It's not yet obvious to me how putting these two together guarantees that you can identify the causal effect? In other words, what happens to the transformation?

---

> > > ### Author Response · Authors · 2023-11-21
> > >
> > > Thanks a lot for the follow-up discussion.
> > >
> > > We would like to clarify that the conditional front-door adjustment variable learned, i.e. Z_CFD, by our method is not a transformed variable. Rather, it is the Z_CFD itself in the assumed data generation mechanism (represented by the DAG  in Figure 2) that our method aims to learn.
> > >
> > > Regarding your concern about the 'transformation', we guess it might come from Equation (7) and the text that follows, which indicates that $X=f(Z_{CFD})+\epsilon$, and this equation or the function $f$ might have led you to consider that Z_CFD is transformed. We would like to clarify that Equation (7) and f as an injective function is an assumption made to prove that our method can identify/learn Z_CFD from data.
> > >
> > > Hope the above explanation helps, and please let us know if you have any further concerns. We really appreciate your comments and discussions.

---

> > > > ### Comment · Reviewer_BvSP · 2023-11-21
> > > > **Follow-up**
> > > >
> > > > To be specific, my concern comes from the last sentence of Theorem 4, which states that the model is A-identifiable, where A represents the transformation up to which the model is identifiable. I understand this so that Theorem 4 does not guarantee that you can identify the true $Z_{CFD}$, only that you can identify some latent variable that is equivalent to the true $Z_{CFD}$ up to the transformation. It is then unclear whether such a transformed variable, when applied in Theorem 3, will lead to the correct causal effect estimate.
> > > >
> > > > Anyway, I will discuss this with the other reviewers and maybe they are able to clarify this to me. If so, I will reconsider my evaluation.
> > > >
> > > > Thanks!

---

> ### Author Response · Authors · 2023-11-22
>
> Thanks for the comment. Now we understand your concern, that is, the learned representation Z_CFD is a transformed version of the true conditional front door adjustment variable. However, since the transformation is linear, i.e., a linear invertible transformation (based on the assumption of iVAE [1]), this transformed variable is still a proper conditional front door adjustment variable because the causal relationships are maintained in this case.
>
> Hope this addresses your concern. Thanks a lot.
>
> [1] Khemakhem, Ilyes, et al. "Variational autoencoders and nonlinear ica: A unifying framework." International Conference on Artificial Intelligence and Statistics. PMLR, 2020.

---

> > ### Comment · Reviewer_BvSP · 2023-11-22
> >
> > "this transformed variable is still a proper conditional front door adjustment variable because the causal relationships are maintained in this case."
> >
> > I understand this so that if you plug-in the transformed variable in Theorem 3, the transformation would cancel and the result would be the same. This sounds plausible, but I would like to see the math. If this is obvious, it shouldn't be too difficult to derive.

---

> > > ### Author Response · Authors · 2023-11-23
> > >
> > > Thanks for your valuable suggestions.
> > >
> > > Please find below the formal analysis to show that using the learned (transformed) CFD adjustment variable will give us the same ATE as using the ground truth CFD adjustment variable.
> > >
> > > This analysis has been added to Appendix B.3 in the latest paper. We have also updated the conclusion to include future work to explore the nonlinear cases.
> > >
> > > Let us consider the linear causal relationships where $ T $ affects $ Y $ through the (conditional) front-door adjustment variable $ Z_{\\rm CFD} $, as represented as follows:
> > >
> > > $Z_{\\rm CFD} = \\alpha T + \\epsilon_{Z_{\\rm CFD}}$ [24]
> > >
> > > $Y = \\beta Z_{\\rm CFD} + \\epsilon_Y$ [25]
> > >
> > > where $ \\epsilon_{Z_{\\rm CFD}} $ and $ \\epsilon_Y $ are the error terms, assumed to be mean-zero, independent random errors. The causal effect of $T$ on $Y$ is commonly computed as the product of the path coefficients, i.e., $ \\text{ATE} = \\alpha \\beta $.
> > >
> > > Following the discussion in Section 4.3, our proposed method CFDiVAE guarantees the model identifiability, which means that the learned latent CFD adjustment variable $\\widetilde{Z_{\text{CFD}}} = {\\widetilde{f}^{-1}}(X)$  is equal to the ground truth CFD adjustment variable ${Z_{\text{CFD}}} = {{f}^{-1}}(X)$, up to a linear invertible transformation (through an invertible matrix $A$) and point-wise nonlinearities (in the form of $S$ and $\\widetilde{S}$), that is,
> > >
> > > $ Z_{\text{CFD}} = A \widetilde{Z}_{\text{CFD}} $  [26]
> > >
> > > Substituting the above expression for $\\widetilde{{Z}_{\\rm CFD}} $ into Eq. 24, we get,
> > >
> > > $ A\\widetilde{Z_{\text{CFD}}} = \\alpha T + \\epsilon_{Z_{\\rm CFD}} $  [27]
> > >
> > > Multiplying both sides of the Eq. 27 by ${A}^{-1}$, we get,
> > >
> > > ${A}^{-1} A\\widetilde{Z_{\text{CFD}}} = \\widetilde{Z_{\text{CFD}}} = \\alpha {A}^{-1}  T + {A}^{-1} \\epsilon_{Z_{\text{CFD}}}$ [28]
> > >
> > > Substituting the expression for $ \\widetilde{Z_{\text{CFD}}} $ shown in Eq. 26 into Eq. 25, we get,
> > >
> > > $Y = \\beta A \\widetilde{Z_{\text{CFD}}} + \\epsilon_Y$ [29]
> > >
> > > Eq. 28 shows that the causal effect of $T$ on $\\widetilde{Z_{\text{CFD}}}$ is $\\alpha {A}^{-1}$, and Eq. 29 shows that the causal effct of $\\widetilde{Z_{\text{CFD}}}$ on $Y$ is $\\beta A$, so the causal effect of $T$ on $Y$ via $\\widetilde{Z_{\text{CFD}}}$ is the product of the two coefficients, i.e., $\\alpha {A}^{-1} \\beta A=\\alpha\\beta {A}^{-1} A=\\alpha\\beta$. This means that using the learned (transformed) CFD adjustment variable will give us the same ATE as using the groundtruth CFD adjustment variable.
> > >
> > > Thanks again for all your valuable comments and suggestions to help us improve our work. We hope our responses address your concerns and help you reconsider your evaluation of our paper.

---

### Official Review · Reviewer_s6a3 · 2023-10-31

**Soundness:** 2 fair
**Presentation:** 3 good
**Contribution:** 3 good
**Rating:** 6
**Confidence:** 4

**Summary:**

**Post rebuttal update** The final version is much better than the original submission. I **raise the score to 6, conditional on** that the following point is fixed in the (possible) camera-ready version. *I believe one main concern of reviewer BvSP is also related to this.*

>The exponential family prior eq11 is too general to be useful. Examine the identifiability in eq25 again, the sufficient statistics S are involved. Only in some cases like Gaussians, we can get around this problem. See [1] and reference therein.

For further improvement, my concern about the outcome regression on the Adult dataset was that here the proxies also affect the outcome. That was why I am asking “is the process in Sec 5 still applicable” for the Adult dataset?

**End update**

The paper considers an extension of the front-door adjustment (FDA), called conditional front-door (CFD) adjustment, where there is a (conditional) variable that affects the adjustment variable in the FDA. An identifiable VAE is employed to recover the adjustment variable from the proxy. Directly based on previous results, the identifiability and adjustment equation of CFD is proved, and the identifiability of the proposed VAE is also proved. Experiments show favorable results.

**Strengths:**

The front-door adjustment (FDA) is an important and understudied approach in the ML community.

Considering a conditional variable in front-door is a meaningful extension.

Using iVAE to recover the FDA variable is novel and interesting.

**Weaknesses:**

*It is very hard to recover the adjustment variable from the proxy (if the proxy noise is not very small)*

The proxy setting is challenging in itself, and the only rigorous identification result under this setting I know is (Miao et al 2018), which depends on two types of proxies and has several assumptions about the proxies.  ref Miao, Wang, Zhi Geng, and Eric J. Tchetgen Tchetgen. "Identifying causal effects with proxy variables of an unmeasured confounder." Biometrika 105.4 (2018): 987-993.

In fact, the current theory is only meaningful under small proxy noise that is \epsilon in eq10.  If we examine the identifiability of iVAE closely, we see, as in eq25 in the current paper, only $f^{-1}(X)$ is identified (up to deterministic transformation) but not the latent variable. And, $f^{-1}(X)=f^{-1}(f(Z)+\epsilon) \neq Z$ if $\epsilon \neq 0$. See the Questions for more related points.

However, if $\epsilon = 0$, then X and Z are related by an injective function, and it is not very meaningful to say X is a proxy of Z, because the whole point of proxy setting is that there is *unmeasured* information of Z not captured in X.

I do not require to fix this weakness in the rebuttal, but it should be discussed and made explicit in the revised version.

*Separated, limited, and unclear outcome estimation*

The connection between the front-door setting and the proposed VAE is weakened, because the outcome is not modeled by the VAE, and the outcome estimation is done in a standalone step, as in Sec 5. Importantly, how does the 2-step process here implement the adjustment eq3? How exactly the ATE is computed? Moreover, the adjustment and outcome variables follow linear models; this might not be an inherent limitation in the approach, but this limitation in the current work should be mentioned in the Conclusion. Finally, for such an important component of the approach, citing a web article makes me nervous, please find and cite an original reference (a research paper).

*Experiments are not sufficient or clear.*

The major problem is that there are no comparisons to methods designed for the front-door setting. And this renders the good result in Fig 4 not very meaningful. I cannot see why we need to exclude FINDFDSET and LISTFDSETS because they require a known DAG; when you build the synthetic dataset, the DAG is known anyway? And I doubt there are no other methods for the FDA. If you really cannot fix this weakness, it is necessary to mention this in the Conclusion.

Another important issue is that, as in Table 1, the proposed method requires a huge amount of data to perform well. In previous work, the methods usually train on sample size < 1k, but the proposed method is worse than other methods when sample size < 1k. This should be examined seriously and would be a major weakness of the approach if this is a general observation.

The experiment on Adult is very confusing to me. First, the “proxy” X also affects Y as shown in the Appendix, can we still say this is a “proxy”? In particular, is the process in Sec 5 still applicable? Second, a reader cannot understand the discussion of direct and indirect causal effects without looking into the Appendix. You need to mention X (eg., marital status) also affects Y and is the “mediator” here. Third, it seems to me that “significant indirect discrimination … through the indirect paths via marital status” and “significant discrimination against sex through the stereotype” are contradictory but not consistent?

*I will read the rebuttal and revised paper and raise my score to 6 if the issues/questions in Weaknesses are addressed. Some further points for improvement are as below.*

**Questions:**

The exponential family prior eq11 is too general to be useful. Examine the identifiability in eq25 again, the sufficient statistics S are involved. Only in some cases like Gaussians, we can get around this problem. See [1] and reference therein.

As is observed in [1], setting the representation dim(Z) to dim(X) is better than dim(Z)=1, and sometimes even better than using the true dim of the adjustment variable. It is better to add experiments in this regard.

*An important reference is missing*. Intact-VAE [1] is also based on iVAE and it is more related to the proposed VAE than any other compared methods, although it considers a different causal setting. In fact, many points mentioned in this review are discussed in [1], e.g., the noise level, the form of the identifiability, and the representation dim. And it is good to add experimental comparison.

[1] Wu, Pengzhou Abel, and Kenji Fukumizu. "beta-Intact-VAE: Identifying and Estimating Causal Effects under Limited Overlap." International Conference on Learning Representations (2022).

I believe only the link W → Z is necessary but not W→T and W→Y. In fact, the Adult dataset does not have W→T.

It should be made clearer in Contributions and Conclusion that it is the FDA, which is a well-established result, that deals with unobserved confounding in this approach, and the VAE learns the adjustment variable but does not do the adjustment using eq3. The CFD is a slight extension of the original FDA.

*Some unsupported/misleading statements*.

“it is often impossible for a CFD variable to be given in practice” This seems strong and needs a reference/discussion.

“there exist unobserved confounding variables and the standard front-door adjustment is no longer applicable” This reads as if FDA does not deal with unobserved confounding but the proposed VAE does.

”the latent variable (i.e., ZCFD) refers to the variable that is not measured, but its information is captured by its proxy” As mentioned, a proxy does not capture all the information of the latent variable, though it might capture *enough* information for the causal effect; this requires challenging analysis and specific assumptions as in Miao 2018.

---

> ### Author Response · Authors · 2023-11-18
>
> Thanks for your review and insights.
>
> Regarding your first weakness about the difficulty of recovering the adjustment variable from its proxies, we have expanded the discussion in the latest version of the paper and included references [3,4] as you suggested.
>
> For your second weakness, we partially concur. CFDiVAE is for learning the representation of the CFD adjustment variable, then after Z_CFD has been learned, we can use it (as the adjustment variable) with any ATE estimation method and the choice of the ATE estimation method is based on the data generation (e.g., linear, or non-linear). For linear models, we propose a basic solution in the main paper. For nonlinear models, we are aware that semi-parametric methods could be applicable, but delving into these methods is outside the scope of our contributions. The initial implementation of our front-door adjustment formula was inspired by reference [1], but we acknowledge your point about its suitability. In the latest version, we now cite both [1] and [2], the latter of which details a semi-parametric method for Natural Indirect Effect (NIE) estimation. It is highly relevant because the causal effect via the front door equates to the NIE in the absence of a direct causal path between T and Y. We have emulated the procedure from [1] and adapted the semi-parametric NIE method to front-door adjustment scenarios. To clarify our approach, we utilize the example causal graph "T->M->Y, T->Y" from mediation analysis, where the NIE represents the effect via T->M->Y, and the Natural Direct Effect (NDE) represents the effect via T->Y. When no causal link exists between T and Y, the NDE becomes zero, and the NIE equates to the Total Effect (TE), making M the front-door adjustment set.
>
> Concerning the Adult dataset, we concur that there is no causal path from W to T, as illustrated in Figure 8. We retain the causal path W to Y based on the comprehensive DAG. In our experimental setup, we categorize all mediating factors under the umbrella of stereotypes, which are regarded as proxies for stereotypes. Thus, the findings on indirect discrimination and discrimination through stereotypes are in alignment. Owing to the scarcity of real-world examples, we can only substantiate the applicability of CFDiVAE in this manner. Following your advice, we will include a discussion on direct and indirect causal effects in the main body of the text.
>
> Following your recommendations, we have incorporated FindFDSet and ListFDsets into our experimental comparison and have stated the limitations of our model, particularly its diminished performance with small sample sizes. We have also revised any unsupported or misleading statements, ensuring that our contributions and conclusions are articulated more clearly.
>
> Please find the latest version in the Rebuttal Revision.
>
> [1] Iain Barr. Causal inference with python part 3 - frontdoor adjustment, Sep 2018. URL http://www.degeneratestate.org/posts/2018/Sep/03/causal-inference-with-python-part-3-frontdoor-adjustment/.
>
> [2] Eric J Tchetgen Tchetgen and Ilya Shpitser. Semiparametric theory for causal mediation analysis: efficiency bounds, multiple robustness, and sensitivity analysis. Annals of statistics, 40(3):1816, 2012.
>
> [3] Wu, Pengzhou Abel, and Kenji Fukumizu. beta-Intact-VAE: Identifying and Estimating Causal Effects under Limited Overlap. International Conference on Learning Representations, 2022.
>
> [4] Miao, Wang, Zhi Geng, and Eric J. Tchetgen Tchetgen. Identifying causal effects with proxy variables of an unmeasured confounder. Biometrika 105.4: 987-993, 2018.

---

> > ### Comment · Reviewer_s6a3 · 2023-11-20
> >
> > Thanks for the rebuttal, and the revised version is indeed better. To push the paper above the bar, things are still needed.
> >
> > **Difficulty of the proxy setting (and others)**
> >
> > The current version reads as if the only requirement is that there are more than 3 proxies. However, if some structural and regularity assumptions are not satisfied, identification is not possible no matter how many proxies there are.
> >
> > Instead, I think you need to stress the importance of *small proxy noise*, since this is in fact how this method can work.
> >
> > It is also needed to restrict the exponential family prior in eq11, as mentioned at the beginning of Questions.
> >
> > **Outcome estimation**
> >
> > This concern remains largely unaddressed. Instead of referring to the NIE paper, I believe you need to discuss the outcome regression used in the paper and answer my questions in the original review. Maybe you could ask the author of the web article for help. At the very least, you need to mention this limitation of linearity in the Limitations.
> >
> > **Experiments**
> >
> > Related to the above point, it is needed to answer: “is the process in Sec 5 still applicable” for the Adult dataset?
> >
> > On the sample size, to say “which is a common limitation of deep learning-based methods” is a bad excuse. Given your evidence, you could say “which is also observed in other VAE methods”. However, the problem is that the other two VAE methods are not for the CF. So, I suggest adding this to the Limitations.

---

> ### Author Response · Authors · 2023-11-22
>
> Thank you for the follow-up discussion and all your suggestions.
> Please find our responses to your latest comments below.
> We have updated the paper based on your latest comment, and the updated contents are highlighted in blue.
>
> Difficulty of the proxy setting (and others): We have revised all the assumptions as you suggested in the latest version. To clarify these assumptions, we summarize them and our problem setting in Section 4.1.
>
> Outcome estimation: As you recommended, we have now provided more detailed explanations to bridge the gap between Equation (3) and the ATE estimation method. In the latest version, we use the discrete data case as an example to demonstrate that Algorithm 1 in Appendix C aligns with Equation (3). Although the solution for continuous data with the linear model does not mirror Equation (3), it is still consistent with the (conditional) front-door adjustment shown in Theorem 3. As we know, the front-door adjustment consists of two back-door adjustments. In our case with continuous variables, these two back-door adjustments are implemented by regression adjustments, as shown in Equation (30) and Equation (31) in Appendix C. Specifically, for the first backdoor adjustment, we use regression adjustment to get the causal effect of $T$ on ${Z_{\text{\rm CFD}}}$ (i.e., $\beta_{T,{Z_{\text{\rm CFD}}}}$) adjusting on $W$, as shown in Equation (30) and Step 1 of the process in Appendix C; for the second backdoor adjustment, we also use regression adjustment as shown in Equation (31) in Appendix C to obtain the causal effect of ${Z_{\text{\rm CFD}}} $ on $Y$ adjusting on $W$ and $U$. However, for this regression adjustment, since $U$ is unobserved, we follow the approach [1-2] to firstly obtain the error $e_{Z_{\text{\rm CFD}}}$, which is independent of $U$, and regress $e_{Z_{\text{\rm CFD}}}$ on $Y$ instead. In this way, we can eliminate the confounding bias caused by $U$. This is shown in Step 2 of the process in Appendix C.
>
> Experiments: Regarding your question about the Adult dataset, we performed discretization on the dataset. After learning the CFD variable via CFDiVAE, we applied Algorithm 1 (as presented in Appendix C) to estimate the ATE. The solution outlined in Algorithm 1 is consistent with Equation (3).
>
> Limitations: Following the comment, we have updated the conclusion to discuss the limitations.
>
> Please let us know if you have any further comments, concerns, or suggestions. We greatly value your feedback and discussion.
>
> [1] Iain Barr. Causal inference with python part 3 - frontdoor adjustment, Sep 2018. URL http://www.degeneratestate.org/posts/2018/Sep/03/causal-inference-with-python-part-3-frontdoor-adjustment/.
>
> [2] Eric J Tchetgen Tchetgen and Ilya Shpitser. Semiparametric theory for causal mediation analysis: efficiency bounds, multiple robustness, and sensitivity analysis. Annals of statistics, 40(3):1816, 2012.

---

### Official Review · Reviewer_A5AY · 2023-11-05

**Soundness:** 3 good
**Presentation:** 3 good
**Contribution:** 3 good
**Rating:** 6
**Confidence:** 3

**Summary:**

In this paper, the authors propose a method to relax some of the restrictions by introducing the concept of conditional front-door (CFD) adjustment and developing the theorem that guarantees the causal effect identifiability of CFD adjustment. The authors further propose CFDiVAE to learn the representation of the CFD adjustment variable directly from data with the identifiable Variational AutoEncoder. They apply CFDiVAE to a real-world dataset to demonstrate its potential application.

**Strengths:**

1. The writing of the paper is good.  The method potentially has broad applications.
2. The authors provide theoretical analyses for the proposed approach.  The reviewer did not check the details due to the review time limit, but it looks sound.

**Weaknesses:**

1. Additional results on real-world datasets and analysis could strengthen the paper.

2. The section references in the supplemental file are confusing.  E.g., the title of section C.4 refers to ‘Section 4.5’ in the main text. However, section 4.5 in the main text does not exist. This causes trouble for the readers.

**Questions:**

The authors could apply the method to additional real-world datasets, conduct a comparison with existing methods, and provide the analysis to strengthen the paper.  The author also needs to address the errors in the section reference.

---

> ### Author Response · Authors · 2023-11-18
>
> Thanks for your review and insights. We revised the typo in the latest version. Following your comments, we updated more explanations for the real-world application, added two comparison methods based on the standard front door adjustment, and conducted additional experiments on dimensionality analysis. Please find the latest version in the Rebuttal Revision.

---

> > ### Comment · Reviewer_A5AY · 2023-11-23
> >
> > The conditional front-door adjustment is novel and interesting. I'd like to keep the score unchanged.

---

> > > ### Author Response · Authors · 2023-11-23
> > >
> > > Thank you very much for your constructive comments and positive feedback on our work. We are delighted to hear that you find the concept of conditional front-door adjustment both novel and interesting. Your acknowledgment is greatly appreciated and serves as a valuable encouragement to us.

---

### Official Review · Reviewer_ysmt · 2023-11-11

**Soundness:** 3 good
**Presentation:** 4 excellent
**Contribution:** 2 fair
**Rating:** 5
**Confidence:** 4

**Summary:**

This paper proposes a new method of Average Treatment Effect estimation and representation learning through the Conditional Front-Door Criterion (CFD) combined with a variational autoencoder model (CFDiVAE).

The authors start with the standard front-door criterion scenario and relax it by allowing backdoor paths between the treatment and the mediator as well as between the mediator and the outcome. The second relaxation is allowing unmeasured confounding variables between treatment and outcome. A background on the front- and back-door criteria is given after which the authors proceed to state the first contribution of the paper, the Conditional Front-Door adjustment (CFD). The CFD is the same formula as the standard front-door adjustment but with additional conditioning on W (the backdoor paths between the treatment and the mediator). Subsequently, the second contribution of the paper, the CFDiVAE model, is described. It follows existing generative VAE work (such as (Louizos et al.)) and adjusts it to the assumed generative model of CFD. CFDiVAE learns a representation of the mediator, given observed back-door paths (confounding variables) between the treatment and the mediator, and allowing for potential unobserved confounding variables. A thorough analysis of model indentifiability is provided. The paper concludes with a series of experiments. First, estimation bias of ATE is compared to competing causal-VAE methods. CFDiVAE beats the other methods for large enough dataset sizes. Secondly, a short assessment of CFDiVAE on a real dataset is performed.

**Strengths:**

The paper tackles the important problem of causal inference with observational data. It is well-written and easy to follow. The methods the paper builds on (front-door criterion) are useful in many areas, from model interpretability to algorithmic fairness. The first contribution (CFD) is a correct relaxation of the front-door criterion, while the second one (CFDiVAE), while building on existing VAE methods, is well analyzed. . The experiments show that, assuming a linear generative model with a mediator, CFDiVAE estimates ATE better than methods that do not model a mediator directly.

**Weaknesses:**

The main weakness of the paper is its limited novelty.

The main contribution, the Conditional Front-Door Adjustment is a straightforward extension of the front-door criterion (where everything is conditioned on the backdoor path W between the treatment and the mediator; W is assumed to be observed). It, in fact, boils down to plugging in the front door criterion into the backdoor criterion.

One could argue that any concrete DAG which allows for the relaxation of the front door criterion with the do calculus (as mentioned in Pearl, Causality, 2009, page 83, Figure 3.1) can form a basis for a similar model.

The second contribution, the CFDiVAE generative model, builds heavily on similar models present in literature. The generative and inference part follow standard VAE procedures adjusted to the assumed underlying SEM.

While the specific combination of CFD with a generative VAE-based model is novel, it also does seem incremental.

**Questions:**

Would considering relaxations of the front door criterion as mentioned in Pearl, Causality, 2009, page 83, Figure 3.1, result in any major changes to the model (I assume the inference model might have to be adjusted)?

---

> ### Author Response · Authors · 2023-11-18
>
> Thanks for your review and insights.
>
> Regarding your concern on the main contribution, while Pearl's book has informally mentioned the relaxation of the front-door criterion, it does not offer a formalisation or solution to the problem. Our paper not only clearly defines the conditional front-door criterion (CFD) but also provides comprehensive proof to support CFD adjustment, thus making a novel and non-trivial theoretical contribution to causal effect identification in the presence of latent confounders. Additionally, it is important to note that the scarcity of literature on the front-door criterion has resulted in less attention to this potent technique for obtaining unbiased causal effect estimation in the presence of unobserved confounders, and the strict requirements of the standard front-door criterion have restricted its practical use, our paper, therefore, makes a very useful contribution by considering a more realistic scenario to enable the practical use of the technique.
>
> As for the second contribution, a main novelty of our work, as you mentioned, lies in the combination of CFD and the VAE based model, especially the employment of the VAE technique makes it possible to learn a CFD adjustment variable directly from data, enabling data-driven causal effect estimation. Moreover, we employ an identifiable VAE, which theoretically ensures the identifiability of the representation learning process. This represents a novel contribution to the practical application of deep generative models for causal inference.
>
> For the question on the changes to our model when considering the example in Pearl’s book, we agree with you that the inference model would be different given the difference in the causal structure.

---

> > ### Comment · Reviewer_ysmt · 2023-11-23
> >
> > Thank you for your reply and for the revised version of your paper.
> >
> > I agree that papers on the FDA are scarce and this makes them (including yours) interesting.
> >
> > I acknowledge that you provide a formal proof for CFD and formulate it, as far as I know, for the first time, but I maintain that this concept is marginally novel. The proof in Appendix B.1 mirrors the proof of the back-door criterion with the addition of causal calculus on the specific graph you propose needed to get rid of the interventional distributions. In fact, Pearl motivates causal calculus saying that it allows for creating models such as yours. In my opinion, the main appeal of the first contribution (CFD) is the motivation for the particular model with observed "confounders" W and post-treatment proxies X.
> >
> > We are also in agreement regarding the second contribution (adjusting related work on VAE / proxy learning to the proposed graphical model for the CFD). The discussion with a fellow reviewer has led to an improved presentation of the limitations of the borrowed methods.

---

> > > ### Author Response · Authors · 2023-11-23
> > >
> > > Thanks again for all your valuable comments and suggestions to help us improve our work. We hope our latest version has clearly stated the novelty of the contribution and significance of our research problem. We would really appreciate it if you could be generous in reconsidering your evaluation of our paper. Thank you!

---

### Author Response · Authors · 2023-11-18

Dear reviewers and AC,

We sincerely thank all reviewers and AC for their great effort and constructive comments on our work.

As reviewers highlighted, we believe our paper studies an interesting and important problem and provides a novelty and effective method for ATE estimation with the presence of unobserved confounding variables. Reviewers have considered the proposed theorem (Conditional Front-door Adjustment) and method (CFDiVAE) to be both novel and interesting.

In addition, we thank the reviewers for pointing out typos and unclear statements in our paper. We also value the suggestions to compare front-door adjustment-based methods as baselines and to establish a more comprehensive assumption setting for proxy variables. We also appreciate some insights that helped us improve our paper a lot, including adding formal analysis about the validity of the learned CFD adjustment variable and filling the gap between the CFD adjustment and ATE estimation.

In response to these comments, we carefully revised and enhanced our paper with the following additional content:

* **We have revised and re-summarized the assumptions (Section 3), contributions (Sections 1 and 8), limitations (Section 8), and future work (Section 8)** to more clearly state the novelty of the contribution and the significance of our research problem.

* **We have incorporated FindFDSet and ListFDsets into our experimental comparison and updated results in Table 2, Table 3 and Figure 4 in Section 6.** Moreover, **we have clarified more implementation details in the Case Study in Section 6.6.**

* **We have expanded Appendix C with detailed explanations to bridge the gap between Theorem 3 and ATE estimation.** We provided the algorithm for CFD adjustment for ATE estimation with Discrete data and the solution for CFD adjustment for ATE estimation with continuous data under the linear model.

* **We have analysed the validity of the learned (transformed) CFD adjustment variable in Appendix B.3.** The results show that using the learned (transformed) CFD adjustment variable will give us the same ATE as using the ground truth CFD adjustment variable.

For ease of review, these updates have been temporarily highlighted in $\\color{red} \\text{red}$ and $\\color{blue} \\text{blue}$. We hope our response and revision can address all the reviewers' concerns, and we are more than eager to have further discussions with the reviewers in response to these revisions.

Thanks!

Authors of Submission 4638

---

> ### Author Response · Authors · 2023-11-22
>
> Thank you for the follow-up discussion and all your suggestions.
>
> We really appreciate your valuable comments.
>
> We have updated the paper based on the follow-up discussion, and the updated contents are highlighted in blue.

---

### Meta-Review · Area_Chair_aXAw · 2023-12-07

**Metareview:**

The paper introduces a VAE-based approach for causal effect estimation with hidden mediators that could be used as a conditional front-door. Using the identifiable VAE in this context can be a novel enough contribution, particularly considering the experimental results.

However, I would like to point out the following:

* Unless I'm missing something subtle, Theorem 3 is just an application of the do-calculus and can be automatically solved with Tian and Pearl's ID algorithm. The soundness and completeness of the ID algorithm is not an obscure result, it's one of the cornerstones of modern causal inference, and it's a disservice to the ICLR audience not to mention that. As a matter of fact, we can just run it in publicly available tools like http://causalfusion.net or the `causaleffect` package (https://cran.r-project.org/web/packages/causaleffect/vignettes/causaleffect.pdf). I'm dubious of the value of presenting the steps in a proof that is not clearly more intuitive than what an automated theorem-proving tool can do.

* There are other papers that consider the same hidden conditional front door adjustments setup. Some examples are "The Proximal ID Algorithm" (https://arxiv.org/abs/2108.06818), "Causal Inference with Hidden Mediators" (https://arxiv.org/abs/2111.02927) and "Partial Identification of Causal Effects Using Proxy Variables" (https://arxiv.org/abs/2304.04374).

As I mentioned before, the use of identifiable VAE, the associated algorithms and experiments is valuable in itself (particularly for the ICLR crowd), but in light of the above it non-trivially diminishes the novel aspects of the contribution and that should be acknowledged.

**Justification For Why Not Higher Score:**

The novelty is less than what is originally stated in the paper.

**Justification For Why Not Lower Score:**

Previous work doesn't have much of an integration with deep learning, and there is something to be said about discussing this problem in setting where the ICLR crowd is more likely to appreciate.

---

### Decision · Program_Chairs · 2024-01-16

Accept (poster)